# A new analytical method for stability analysis of rock blocks with cavities in sub-horizontal strata by the considering eccentricity effect

Xushan Shi[1], Bo Chai[1,3,4], Juan Du[1,2,4], Wei Wang[1,3,4], Bo Liu[1]

[1]School of Environmental Studies, China University of Geosciences, Wuhan, 430078, China
[2]Centre for severe weather and climate and hydrogeological hazards, Wuhan, 430078, China
[3]Hubei Key Laboratory of Yangtze Catchment Environmental Aquatic Science, Wuhan, 430078, China
[4]Research Center for Geohazards Monitoring and Warning in Three Gorges Reservoir, Wanzhou, 404000, China

*Correspondence to*: Bo Chai (chaibo@cug.edu.cn) and Juan Du (dujuan@cug.edu.cn)

**Abstract.** The basal cavity of a rock block formed due to differential weathering is an important predisposing factor for rockfall in hard-soft interbedded rocks, which induces eccentricity effect at the base of the rock block. Rock block falling due to the eccentricity effect with the failure modes of toppling or sliding is defined as biased rockfall in this study. Taking into account the non-uniform stress distribution due to the eccentricity effect, a new analytical method is proposed for three-dimensional stability force and stability of biased rockfall. The development of non-uniform stress distribution stress calculated by this analytical method was verified by numerical simulation. The biased rockfall progresses from partial damage of the soft underlying layer, caused by non-uniform distributed stress, to toppling and sliding of overhanging hard rock block due to overall unbalanced force. Therefore, a set of factors of safety ($Fos$) against partial damage (compressive and tensile damage of the soft underlying layer) and overall failure (toppling and sliding of the hard rock block) are used to determine the rockfall susceptibility level. The analytical method is applied and validated using biased rockfalls on the northeast edge of the Sichuan Basin in Southwest China, where a significant number of rockfalls consisting of overhanging thick sandstone and underlying mudstone occur.overhanging The evolution process of biased rockfalls is divided into four stages, initial state, cavity formation, partial unstable and failure. The proposed method is validated by calculating $Fos$ of the typical unstable rock blocks in the study area. As the cavity continues to grow, the continuous retreat of cavity causes stress redistribution between the hard and soft rock layers. This results in damage to the underlying soft rock layer due to the development of the eccentricity effect, ultimately leading to the failure of the hard rock block. The critical cavity retreat ratio is determined to be 0.33, which is used to classify the low and moderate rockfall susceptibility in the eastern Sichuan Basin. The proposed analytical method provides insights into the evolution of biased rockfall and a means for early identification and susceptibility assessment of rockfall.

**List of symbols**

| | |
|---|---|
| $a$ | length of the block along the $x$ direction |
| $A$ | area of contact surfaces |
| $b$ | width of the block along the $y$ direction |
| $c$ | cohesive force of the mudstone |

| 32 | $d_i$ | width of the basal cavity in a certain direction |
|---|---|---|
| 33 | $e_x$ | eccentric distance along the $x$ direction |
| 34 | $e_y$ | eccentric distance along the $y$ direction |
| 35 | $E_x$ | horizontal seismic force along the $x$ direction |
| 36 | $Fos$ | factor of safety |
| 37 | $h$ | height of the block |
| 38 | $h_w$ | height of the water in the fracture |
| 39 | $H_x$ | water pressure along the $x$ direction |
| 40 | $I_x$ | moment of inertia with respect to the $x$-axis |
| 41 | $I_y$ | moment of inertia with respect to the $y$-axis |
| 42 | $k_e$ | earthquake contribution coefficient |
| 43 | $k_1$ | rainfall coefficient, taking 1 in the rainfall scenario and 0 in the non-rainfall scenario |
| 44 | $k_2$ | earthquake coefficient, taking 1 in the seismic scenario and 0 in the non-seismic scenario |
| 45 | $k_3$ | free surface coefficient, taking 1 for two free surfaces and 0 for three free surfaces |
| 46 | $M_{bx}$ | total bending moments with respect to the $x$-axis on the mudstone foundation |
| 47 | $M_{by}$ | total bending moments with respect to the $y$-axis on the mudstone foundation |
| 48 | $M_{bEx}$ | bending moment of $E_x$ with respect to the $x$-axis on the mudstone foundation |
| 49 | $M_{bHx}$ | bending moment of $H_x$ with respect to the $x$-axis on the mudstone foundation |
| 50 | $M_{bWx}$ | bending moment of $W$ with respect to the $x$-axis on the mudstone foundation |
| 51 | $M_{Ex}$ | overturning moment provided by $E_x$ along the $x$ direction |
| 52 | $M_{Hx}$ | overturning moment provided by $H_x$ along the $x$ direction |
| 53 | $M_{px}$ | stabilizing moment of $p_n$ along the $x$ direction |
| 54 | $M_{W_{inx}}$ | stabilizing moment provided by $W$ along the $x$ direction |
| 55 | $M_{W_{outx}}$ | overturning moment provided by $W$ along the $x$ direction |
| 56 | $N_z$ | total applied vertical load on the mudstone base |
| 57 | $O$ | origin of the $(x, y)$ coordinates |
| 58 | $p(x, y)$ | pressure magnitude at point $(x, y)$ |
| 59 | $r_i$ | the basal cavity retreat ratio equal to the ratio of cavity width to block width in a certain direction |
| 60 | $W$ | weight of the block |
| 61 | $x$ | distance to $O$ along the $x$-axis |
| 62 | $y$ | distance to $O$ along the $y$-axis |
| 63 | $\alpha$ | true dip of the contact surface |
| 64 | $\gamma_s$ | unit weight of sandstone |

| 65 | $\gamma_w$ | unit weight of water |
|----|-----------|----------------------|
| 66 | $\theta_1$ | apparent dip of $\alpha$ on plane J1 |
| 67 | $\theta_2$ | apparent dip of $\alpha$ on plane J2 |
| 68 | $\sigma_{cmax}$ | ultimate compressive strength of the mudstone |
| 69 | $\sigma_{tmax}$ | ultimate tensile strength of the mudstone |
| 70 | $\tau_{max}$ | ultimate shear strength of the mudstone |
| 71 | $\varphi$ | friction angle of the mudstone |
| 72 | $\omega_1$ | angle between the trend of the contact surface and the $x$ direction |
| 73 | $\omega_2$ | angle between the trend of the contact surface and the $y$ direction |

## 1 Introduction

Rockfall is defined as the detachment of a rock block from a steep slope along a surface, on which little or no shear
displacement takes place (Cruden and Varnes, 1996). Rockfalls frequently occur in mountainous ranges, cut slopes, and coastal
cliffs, and they may cause significant facility damage and casualties in residential areas and transport corridors (Chau et al.,
2003; Volkwein et al., 2011; Corominas et al., 2018). Stability analysis of rock blocks are crucial for risk management and
early warning of rockfall (Kromer et al., 2017).
Rockfalls are prone to occur in soft-hard rock formations, and the non-uniform stress distribution caused by differential
weathering of rock formations is the main reason for the failure of rockfall. In the eastern Sichuan Basin, Southwest China,
rockfall is widespread and poses high risk (Chen et al., 2008; Chen and Tang, 2010; Zhang et al., 2016; Zhou et al., 2017;
Zhou et al., 2018). The rockfall in this area is attributed to the tectonic setting of Jura-type folds and the stratum sequence,
which is characterized by the interbedding of hard and soft layers. An alternation of thick sandstone and thin mudstone layers
is formed in the wide and gentle-angle synclines (Zhang et al., 2015; Wu et al., 2018). Weathering is known to be one of the
main predisposing factors for rockfall (Jaboyedoff et al., 2021; Zhan et al., 2022). The cliff comprised of hard sandstone is the
source of rockfall, and the underlying mudstone is more susceptible to weathering. Along with the retreat of basal cavities in
the mudstone layer, the gravity centre of the overhanging sandstone block moves outward relative to the mudstone. In this
case, the stress distribution in the contact surface of sandstone and mudstone is non-uniform. The mudstone on the outer side
bears higher compressive stress than that on the inner side. This phenomenon can be defined as an eccentricity effect, which
leads to mudstone damage and failure of the overhanging sandstone by toppling or sliding. This type of rockfall is defined as
biased rockfall in this study (Fig. 1). Similar rockfall patterns have been widely reported in other regions, such as Joss Bay in
England (Hutchinson, 1972), Okinawa Island in Japan (Kogure et al., 2006), and the Colorado Plateau of the southwestern
United States (Ward et al., 2011). Retreat of the basal cavity is a main cause for the failure of the overhanging block. Therefore,
it is necessary to establish an analytical method, considering the development of the basal cavity, to analyse the stress

distribution and stability of rock blocks, which is fundamental to the susceptibility assessment and risk control of biased rockfall.

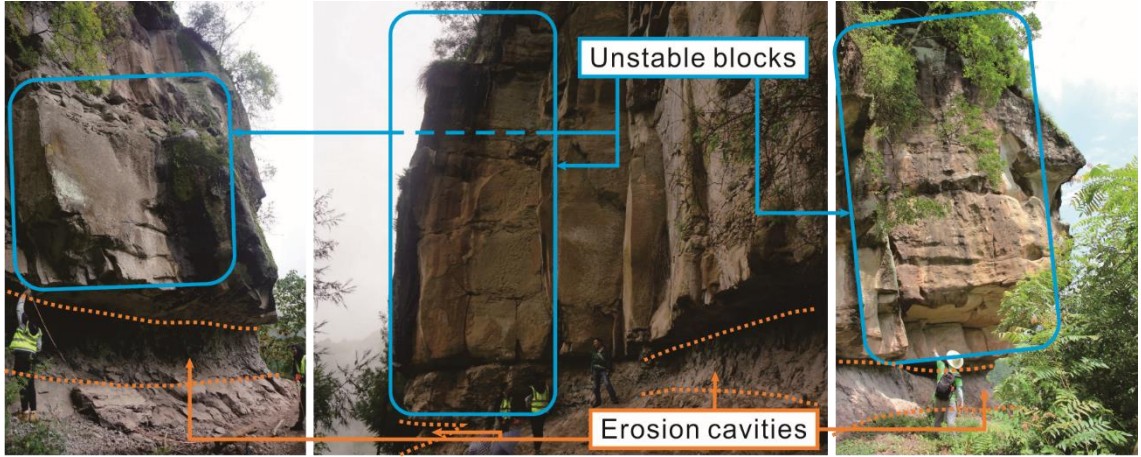

**Figure 1** Potential unstable blocks and basal cavities caused by differential weathering.

Rockfall stability analysis methods include statistical analysis (Frattini et al., 2008; Santi et al., 2009), empirical rating systems (Pierson et al., 1990; Ferrari et al., 2016), and mechanical analysis (Jaboyedoff et al., 2004; Derron et al., 2005; Matasci et al., 2018). The statistical analysis and empirical rating systems are suitable for rockfall hazard assessment at a regional scale. The accuracy of statistical analysis depends on the completeness of rockfall inventories (Chau et al., 2003; Guzzetti et al., 2003; D'amato et al., 2016). However, its application to rockfall hazards is limited due to the lack of complete inventory data (Budetta and Nappi, 2013; Malamud et al., 2004). Empirical and semi-empirical rating systems are used where site-specific rockfall inventories are either unavailable or unreliable. Therefore, rockfall susceptibility can be assessed by heuristic ranking of selected predisposing factors (Frattini et al., 2008; Budetta, 2004). Mechanical analysis based on static equilibrium theory is the main method to analyse the stability of site-specific rockfall using the factor of safety ($Fos$). Ashby (1971) conducted stability analysis with a parallelepiped block resting on an inclined plane (Fig. 2a), and the solution was subsequently modified by Bray and Goodman (1981) and Sagaseta (1986). Kogure et al. (2006) utilized a cantilever beam model to determine the critical state of limestone cliffs. Frayssines and Hantz (2009) proposed the limit equilibrium method (LEM) to predict block stability against sliding and toppling in steep limestone cliffs (Fig. 2c). Chen and Tang (2010) established a stability analysis method of three types of unstable rocks in the Three Gorges Reservoir area with the LEM. Alejano et al. (2015) studied the influence of rounding of block corners on the block stability. Zhang et al. (2016) defined $Fos$ based on fracture mechanics and studied the progressive failure process by analysing crack propagation. Alejano et al. (2010) and Pérez-Rey et al. (2021) deduced a formula for $Fos$ of blocks with more complex geometry.

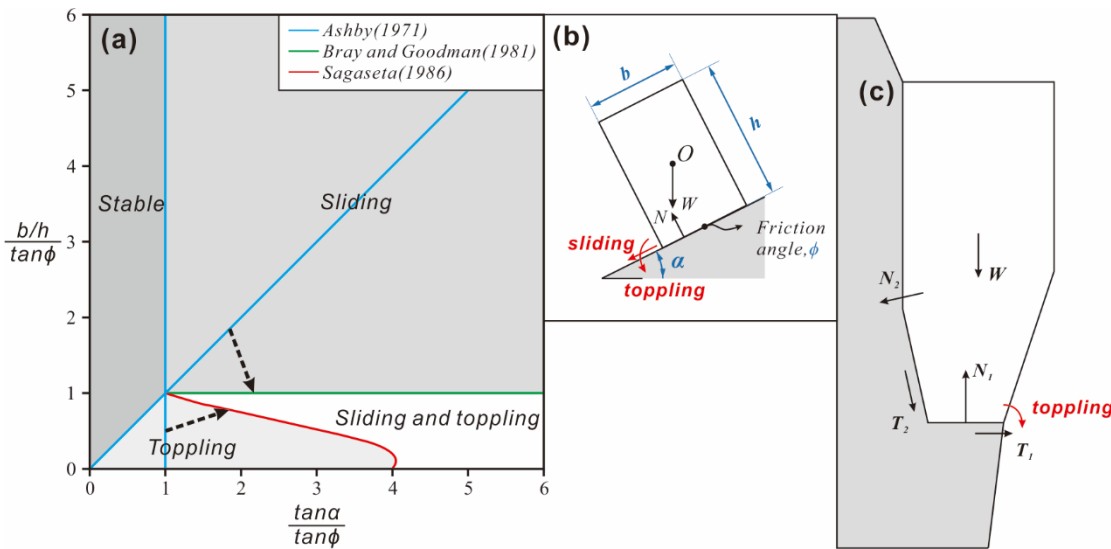


**Figure 2** Traditional force analysis diagrams of the rock block. (a) and (b) are stability analysis diagrams of rock blocks under dynamic
conditions, resting on an inclined plane with a dip angle of α. The rock block is generalized as a cuboid with dimensions $b \times h$ and weight
$W$ (as modified from Ashby (1971), Bray and Goodman (1981) and Sagaseta (1986)). (c) Force description of the toppling model proposed
by Frayssines and Hantz (2009). In the above assumptions, $N$, $T$, and $W$ are regarded as forces applied at a point.
The supporting force on the contact surface is assumed to be applied at a point in the current LEM methods (i.e., N in Fig. 2 b
and c). However, the supporting force is actually a distributed force. The cavity generates an eccentricity effect on the
overhanging rock block and results in a non-uniform distribution of the supporting force on the contact surface, which is not
considered in the traditional LEM. The presence of non-uniform stress distribution plays a critical role in inducing localized
damage within a rock mass. Damage is frequently considered as an indicator or a threshold for the onset of accelerated failure
in rock masses (Zhang et al., 2016). Therefore, it is imperative to consider the non-uniform stress distribution for the rockfall
stability analysis. Furthermore, most studies simplified the three-dimensional geometry of the rock block by one cross-section,
which is used to represent the critical features of the slope structure. Nevertheless, for natural blocks with basal cavities, the
cavities usually present different depths along different directions (Pérez-Rey et al., 2021). Therefore, a three-dimensional
model is necessary to calculate the accurate stability. In addition, when a block has multiple free faces and a complex structure,
its potential failure is dominated by different modes, including rock mass damage and overall block failure. Therefore, the
probable failure modes should be determined prior to the calculation of $Fos$.
Based on rockfall investigation in the Eastern Sichuan Basin, China, the main objective of this study was to propose a new
three-dimensional method for the determination of failure modes and $Fos$ of biased rockfall, considering the non-uniform
force distribution on the contact surfaces. Compared with the traditional LEM method, this study takes into account the partial
damage of the underlying soft rock and the overall instability of the overhanging hard rock blocks, and can evaluate the stability
of biased rockfall more comprehensively. $Fos$ of the typical unstable rock blocks in the study area are calculated to validate
the proposed method. In addition, the critical cavity retreat ratio in this area is analysed. This study is an extension of the basic
LEM for rockfall, which can promote the accuracy of rockfall stability analysis and facilitate rockfall prevention and risk
mitigation.
**2 Study area**
**2.1 Geological setting**
The study area is located on the northeastern edge of the Sichuan Basin, China (Fig. 3a). Continuous erosion processes generate
moderate-low mountain and valley landforms (Yu et al., 2021). The tectonic structure of this area is characterized by a series
of ENE anticlines and synclines (Fig. 3b, c). In the anticline area, the rock layers dip relatively steeply, where translational
rockslides are the main mode of slope failure. The syncline area is dominated by gently dipping strata and is prone to rockfall
(Zhou et al., 2018). The study area is located in the core of the Matouchang syncline, where the rock layers are sub-horizontal
(Fig. 3d, e). In this valley, due to the longstanding fluvial incision, the relative relief is approximately 500 m and the valley
flanks are extremely steep (Fig. 3e). In addition, the toes of the hill slopes are reshaped because of the construction of the
G318 national road, which is the main traffic line and is always threatened by rockfalls dropping from steep rock slopes (shown
in Fig. 3d and Table 1).

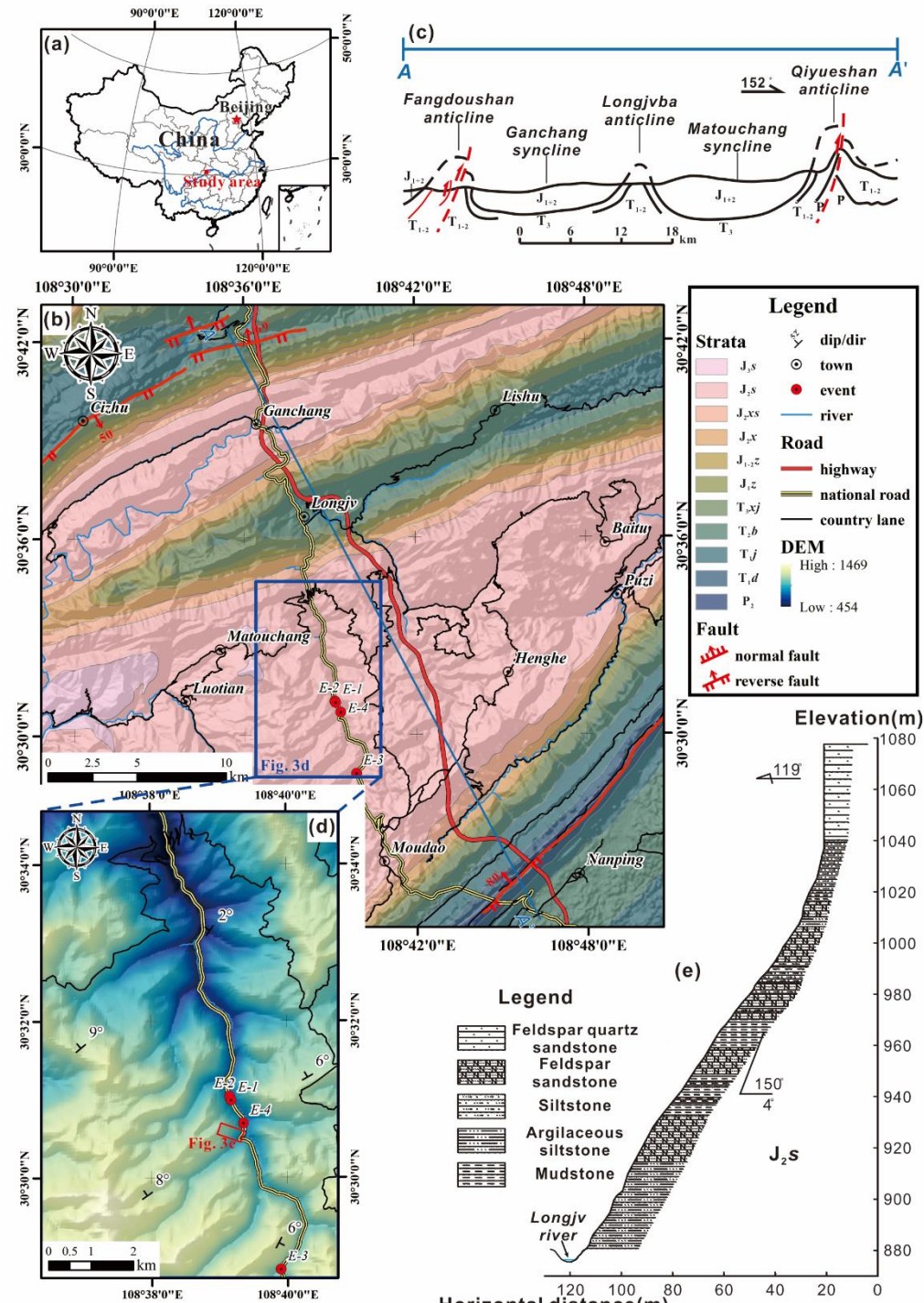


**Figure 3** (a) Location of the study area in China; (b) geological map of the study area; (c) tectonic sketch profile of A-A', whose location is

showed in Fig. 3b; (d) rockfall-prone segment and key investigation areas. The red dots are the positions of historical rockfall events,

 corresponding to the numbers in Table 1; (e) Geological cross-section of the hillslope in the Jitougou section of G318 national road, which
is marked by a red rectangle in Fig. 3d.
**Table 1** Historical rockfall events along G318 national road in the study area

| No | Location | Time of occurrence (GMT+8) | Volume [m³] | Consequence |
|----|----------|----------------------------|-------------|-------------|
| E-1 | K1698+900 | 2014-05 to 06* | Unknown | The power transmission facilities outside the road were smashed. |
| E-2 | K1699+000 | 2015-02-14 23:00 | About 240 | A passing truck was stuck and two people dead. |
| E-3 | K1690+700 | 2015-06-16 | Unknown | The road was interrupted for a day. |
| E-4 | K1698+400 | 2015-06-18 09:00 | About 200 | A vehicle was crashed into a gully and four people dead. |

*Note: The exact time is unknown.

## 2.2 Rockfall characteristics

The slopes in the study area consist of a sub-horizontally interbedded sandstone and mudstone layer. Therefore, there are
multiple layers of potentially unstable rock blocks in the hill slopes (Fig 4a). The thick sandstone has two sets of sub-vertical
joints (Fig. 5), which cut the rock mass into blocks as the potential rockfall source (Fig. 4b). Cavities have formed in the
underlying mudstone layer (Fig. 4c, d). Joints and bedding planes (BP) constitute the detachment surfaces between the blocks
and steep slope (Fig. 4e). The eccentricity effect produced by the mudstone cavity plays an important role in the evolution
process of rockfall. When the basal mudstone cannot provide adequate supporting force, the blocks detach from the steep slope,
and biased rockfall occurs. Sliding and toppling are two possible failure modes of biased rockfall.

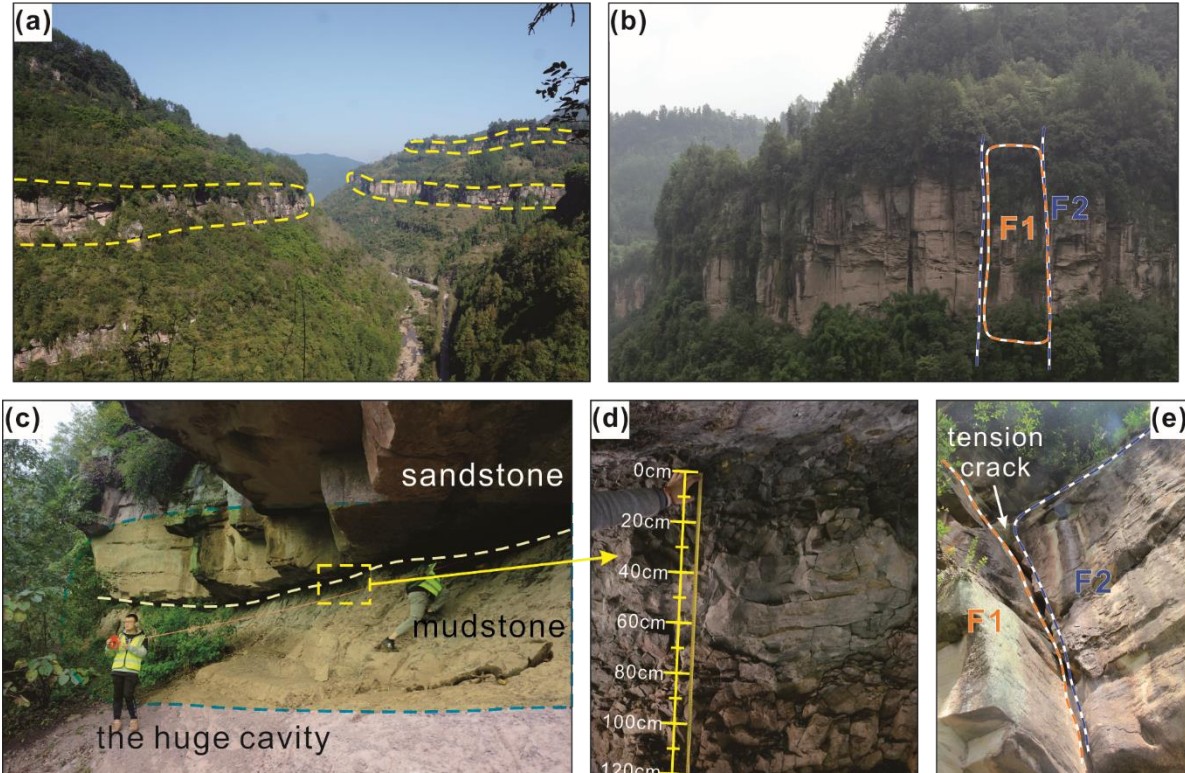


**Figure 4** Characteristics of biased rockfalls in the study area. (a) Multiple-layers of rockfall sources, which is consist of thick sandstone. (b)
Two sets of sub-vertical joints (F1 and F2) recognized by the UAV photos. (c) Large basal cavity developed in the underlying mudstone. (d)
Dense fractures on the mudstone surface generated by weathering and compression. (e) Vertical tension crack in the rear of the block, through
which precipitation can infiltrate.
According to the historical rockfall events in this area, precipitation is considered a triggering effect of rock instability. The
precipitation mainly infiltrates along the sub-vertical joints or cracks of the sandstone (Fig. 4e). However, the drainage of
fissure water is hysteretic due to the obstruction of basal mudstone. Therefore, transient steady flow exists in vertical cracks
during heavy rainfall, and the hydrostatic pressure triggers the detachment of rock blocks. Thus, typical scenarios (such as
rainfall intensity and earthquake) need to be considered in the stability analysis model.

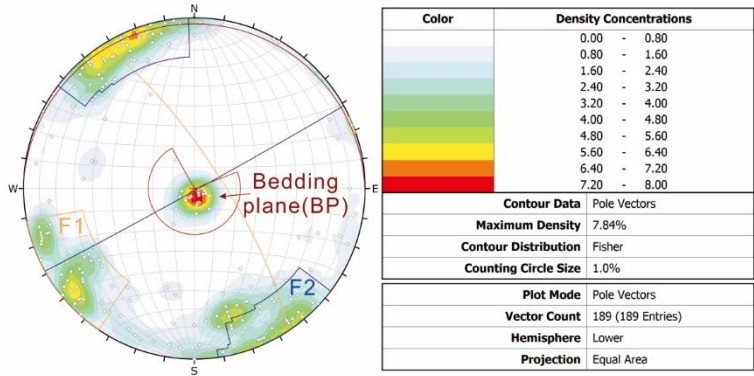

**Figure 5** Stereo net produced using compass-clinometer survey data, which shows the densities and orientations of five clusters. The data
were collected in the rockfall-prone area shown in Fig. 3d.
**3 Calculation method**
**3.1 Geological models and assumptions**
A detailed geological investigation of unstable rock blocks was carried out in the study area (Fig. 6). The geological model of
the rock block is mainly composed of the overhanging sandstone and the underlying mudstone. The sandstone block is assumed
to be a rigid body, which is divided by two sets of orthogonal vertical smooth joints without friction resistance. According to
the relatively persistent sub-vertical fractures observed in the field, the vertical joints are assumed to be fully persistent in the
geological model. The sandstone block is assumed to be a complete body without persistent discontinuity, and it will not
disintegrate before it falls. Due to the cavity in mudstone, the contact surface between sandstone and mudstone exhibits an
eccentricity effect where non-uniform stresses are distributed at different positions. Mudstone is mainly loaded by compressive
stress and tensile stress. When the compressive stress of mudstone exceeds its strength on the outer side, some initial damage
appears. The effective contact surface between mudstone and sandstone is reduced, which aggravates the non-uniform
distribution of stress. In this way, the ability of mudstone to resist the sliding and toppling of overhanging sandstone is reduced.
In the field, compression deformation of mudstone can be observed, which usually manifests as micro-fractures and cleavages
(Fig. 4d). The deformation is very slight and slow in the short term. In addition, the LEM is essentially a force/stress approach
that does not take into account the deformation. Therefore, in this study, it is assumed that the mudstone is not subjected to
deformation. The rock block remains in the state of static equilibrium prior to the final overall failure. Fig. 7 displays the four
evolution stages of biased rockfall. In the initial stage, the base cavity has not yet formed, and the normal force acting on the
contact surface is uniform in different positions. The eccentricity effect leads to a non-uniform supporting force as the cavity
grows, and partial damage gradually develops when the non-uniform stress exceeds the compressive or tensile strength of the
mudstone. Under the triggering effects of rainfall or earthquakes, the rock blocks are separated by sliding or toppling.

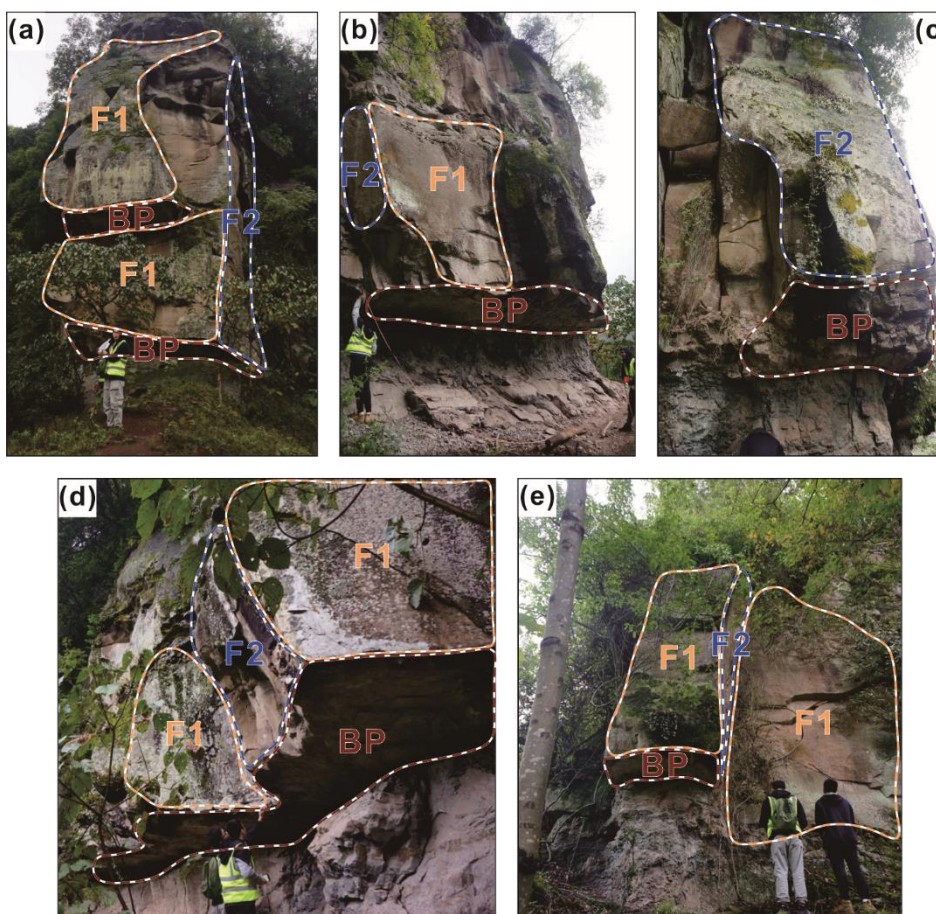


**Figure 6** The unstable blocks were labelled W02, W08, W18, W04, and W21, which are detached by the dominating discontinuities in Fig.
5. Basal cavities can be identified under the bedding planes of sandstone.

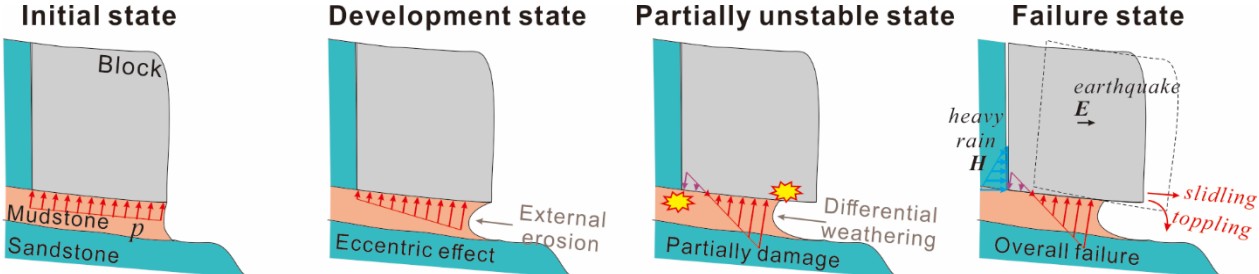

**Figure 7** The evolution process of rock blocks from stable state to failure.

Fig. 8 represents the mechanical model of the force equilibrium analysis of a rock block with two or three free faces. The rock block (the overhanging sandstone) is generalized as a parallelepiped block. The underlying mudstone is impermeable, so rainfall can fill the joints and transmit horizontal hydrostatic pressure. The shear strength of the underlying mudstone is assumed to obey the Mohr–Coulomb criterion. Rainfall and earthquakes decrease $Fos$ by generating hydrostatic pressure $H$ in the vertical crack and horizontal seismic force $E$ on the block.

A Cartesian coordinate system is established in three-dimensional space for the force analysis. The origin $O$ is located at the centre of the contact surface between sandstone and mudstone. For the case with two free surfaces, the orientation of the free surfaces is set to be the positive direction of the $x$-axis and $y$-axis. For the case with three free surfaces, the negative direction of the $x$-axis is also a free surface. Joint J2 is perpendicular to the $x$-axis, and joint J1 is perpendicular to the $y$-axis.

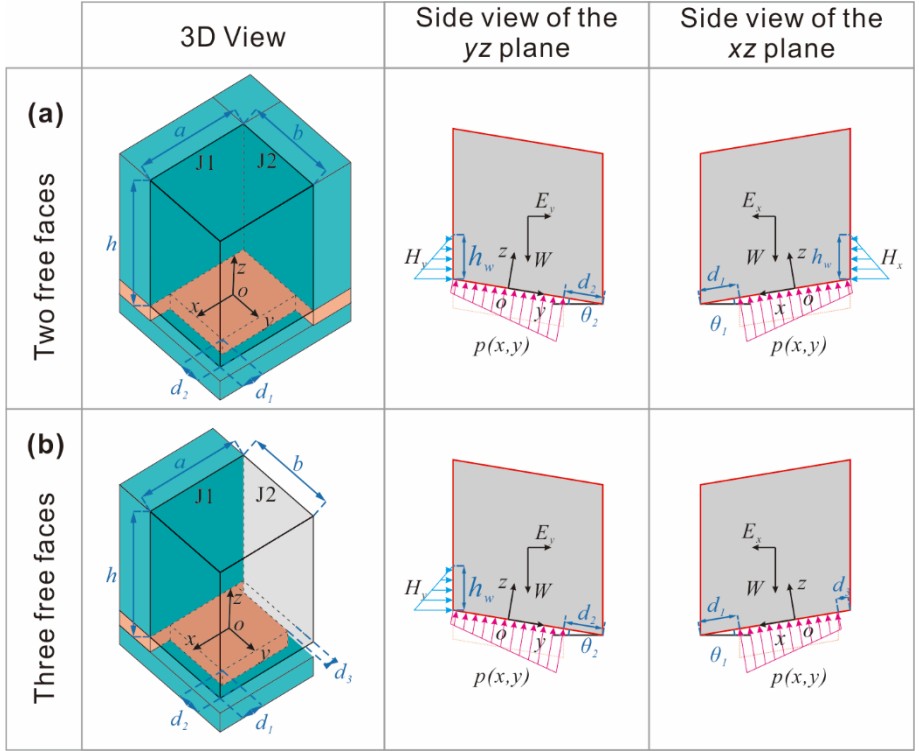

 **Figure 8** Diagram of the force equilibrium analysis of the rock block model. (a) and (b) represent the case of unstable rock blocks with

two or three free vertical surfaces, respectively.
**3.2 Calculation processes**
**3.2.1 Stress distribution at the block base**
The following formulas are used to calculate the apparent dip of $\alpha$ ($\theta_1$ and $\theta_2$):

$$\theta_1 = \arctan(\tan \alpha \cdot \cos \omega_1) \tag{1}$$

$$\theta_2 = \arctan(\tan \alpha \cdot \cos \omega_2) \tag{2}$$

where $\omega_1$ and $\omega_2$ are the angles between the trend of the contact surface and the $x$ direction or $y$ direction, respectively.
As shown in Fig. 8b, with respect to the $x$-axis, gravity, seismic forces, and hydrostatic pressure create a non-symmetrical
stress distribution on the foundation. The bending moment of gravity with respect to the $x$-axis ($M_{bWx}$) is

$$M_{bWx} = W \cdot \frac{d_1 - d_3}{2} \cos \theta_1 \tag{3}$$

Assuming that the height of the water in the fracture is $h_w$, the hydrostatic pressure along the $x$ direction ($H_x$) and its bending
moment ($M_{bHx}$) are respectively expressed as

$$H_x = \frac{\gamma_w h_w^2}{2}(b - d_2) \tag{4}$$

$$M_{bHx} = \int_{-\frac{b-d_2}{2}}^{\frac{b-d_2}{2}} \int_0^{h_w \cos \theta_1} \gamma_w \left( h_w - \frac{z}{\cos \theta_1} \right) \left( \frac{z}{\cos \theta_1} + \frac{a - d_1 - d_3}{2} \cdot \sin \theta_1 \right) dz\, dy \tag{5}$$

The horizontal seismic force along $x$ direction ($E_x$) and its bending moment ($M_{bEx}$) are respectively expressed as

$$E_x = k_e W \tag{6}$$

$$M_{bEx} = E_x \left( \frac{h}{2} - \frac{d_1 - d_3}{2} \sin \theta_1 \right) \tag{7}$$

The total applied vertical load ($N_z$) and the total bending moments along the $x$ direction ($M_{bx}$) can be derived as

$$N_z = W \cos \alpha - (H_x \cdot k_1 \cdot k_3 + E_x \cdot k_2) \sin \theta_1 - (H_y \cdot k_1 + E_y \cdot k_2) \sin \theta_1 \tag{8}$$

$$M_{bx} = M_{bWx} + M_{bHx} \cdot k_1 \cdot k_3 + M_{bEx} \cdot k_2 \tag{9}$$

where $k_1$, $k_2$ and $k_3$ are the coefficients set to make Eq. (8) and Eq. (9) compatible with different calculation scenarios.
Therefore, Eqs. (8) and (9) and the following formulas can be expressed in a unified form. In the natural scenario, $k_1$ and $k_2$
are both equal to 0. In the rainfall scenario, $k_1 = 1$. In the earthquake scenario, $k_2 = 1$. For the case of two free faces, $k_3 = 1$.
For the case of three free surfaces, $k_3 = 0$.
Based on bending theory (Adrian, 2010), the eccentricity distance along the $x$ direction ($e_x$) can be expressed as

$$e_x = \frac{M_{bx}}{N_z} = \frac{M_{bWx} + M_{bHx} \cdot k_1 \cdot k_3 + M_{bEx} \cdot k_2}{W \cos \alpha - (H_x \cdot k_1 \cdot k_3 + E_x \cdot k_2) \sin \theta_1 - (H_y \cdot k_1 + E_y \cdot k_2) \sin \theta_1} \tag{10}$$

The same method can be used to obtain $e_y$:

$$e_y = \frac{M_{by}}{N_z} = \frac{M_{bWy} + M_{bHy} \cdot k_1 + M_{bEy} \cdot k_2}{W \cos \alpha - (H_x \cdot k_1 \cdot k_3 + E_x \cdot k_2) \sin \theta_1 - (H_y \cdot k_1 + E_y \cdot k_2) \sin \theta_1} \quad (11)$$

According to the stress distribution of a rectangular shaped foundation (Adrian, 2010), the stress in the $(x, y)$ coordinates,
$p(x, y)$, is

$$p(x, y) = \frac{N}{A} + \frac{Ne_x}{I_y}x + \frac{Ne_y}{I_x}y \quad (12)$$

with the formulas

$$I_x = \frac{(a - d_1)(b - d_2)^3}{12} \quad (13)$$

$$I_y = \frac{(b - d_2)(a - d_1)^3}{12} \quad (14)$$

$$A = (a - d_1 - d_3)(b - d_2) \quad (15)$$

By substituting Eq. (13-15) into Eq. (12), $p(x, y)$ can be derived as

$$p(x, y) = \frac{N}{A}\left[1 + \frac{12e_x}{(a - d_1 - d_3)^2}x + \frac{12e_y}{(b - d_2)^2}y\right] \quad x \in \left[-\frac{a - d_1 - d_3}{2}, \frac{a - d_1 - d_3}{2}\right], y \in \left[-\frac{b - d_2}{2}, \frac{b - d_2}{2}\right] \quad (16)$$

$p_{max}$ and $p_{mim}$ can be derived from Eq. (16) as

$$p_{max} = p\left(\frac{a - d_1 - d_3}{2}, \frac{b - d_2}{2}\right) \quad (17)$$

$$p_{min} = p\left(-\frac{a - d_1 - d_3}{2}, -\frac{b - d_2}{2}\right) \quad (18)$$

The mudstone foundation has both compressive strength and tensile strength, so the value of $p(x, y)$ is modified to obtain the
two piecewise functions

$$p_p(x, y) = \begin{cases} \sigma_{cmax}, & p(x, y) \geq \sigma_{cmax} \\ p(x, y), & 0 < p(x, y) \leq \sigma_{cmax} \\ 0, & p(x, y) < 0 \end{cases} \quad (19)$$

$$p_n(x, y) = \begin{cases} 0, & p(x, y) < -\sigma_{tmax} \\ p(x, y), & -\sigma_{tmax} \leq p(x, y) < 0 \\ 0, & p(x, y) \geq 0 \end{cases} \quad (20)$$

Here, $p_p(x, y)$ provides support normal force for the overhanging sandstone, and $p_n(x, y)$ provides tension force.
**3.2.2 Calculation of factors of safety**
According to the Mohr-Coulomb criterion, the ultimate shear strength $\tau_{max}$ is

$$\tau_{max} = \int_{-\frac{a-d_1-d_3}{2}}^{\frac{a-d_1-d_3}{2}} \int_{-\frac{b-d_2}{2}}^{\frac{b-d_2}{2}} \left[p_p(x, y) \tan \varphi + c\right] dy \, dx \quad (21)$$

Therefore, $Fos$ against sliding, $Fos_{sl}$, can be defined as
$$Fos_{sl} = \frac{S_{stabilizing}}{S_{sliding}} = \frac{\tau_{max}}{W|\sin\alpha_s| + H_x \cdot \cos\omega_s \cdot \cos\alpha_s \cdot k_1 \cdot k_3 + H_y \cdot |\sin\omega_s| \cdot \cos\alpha_s \cdot k_1 + E \cdot \cos\alpha_s \cdot k_2} \quad (22)$$
When the block can slide freely, $\alpha_s = \alpha$, $\omega_s = 0$; when the block is constrained to slide along a joint plane (e.g., J1), $\alpha_s =$
$\theta_1$ or $\theta_2$, $\omega_s = \omega_1$ or $\omega_2$. For the case of an anaclinal slope, the sliding direction is opposite to the free surface. Therefore, the
rock block does not slide, and $Fos_{sl}$ is not considered in the model.
With regard to stability against toppling, along the $x$ direction, the part of the block above the mudstone base provides the
stabilizing moment $M_{W_{inx}}$, and the part of the block above the cavity provides the overturning moment $M_{W_{outx}}$. When tension
exists, there is an additional stabilizing moment. $M_{px}$, $M_{W_{inx}}$, $M_{W_{outx}}$ and $M_{px}$ can be derived as
$$M_{W_{inx}} = W\frac{a-d_1}{a}\cos\theta_1 \cdot \left(\frac{a-d_1}{2}\right) \quad (23)$$
$$M_{W_{outx}} = W\frac{d_1}{a}\cos\theta_1 \cdot \frac{d_1}{2} \quad (24)$$
$$M_{px} = -\int_{-\frac{b-d_2}{2}}^{\frac{b-d_2}{2}}\int_{-\frac{a-d_1-d_3}{2}}^{\frac{a-d_1-d_3}{2}} p_n(x,y) \cdot \left(\frac{a}{2} - d_1 - x\right) dx\, dy \quad (25)$$
and $M_{Hx}$ and $M_{Ex}$ can be derived as
$$M_{Hx} = \int_{-\frac{b-d_2}{2}}^{\frac{b-d_2}{2}}\int_0^{h_w\cos\theta_1} \gamma_w\left(h_w - \frac{z}{\cos\theta_1}\right)\left(\frac{z}{\cos\theta_1} + (a-d_1)\sin\theta_1\right) dz\, dy \quad (26)$$
$$M_{Ex} = E_x\left(\frac{h}{2} + \left(\frac{a}{2} - d_1\right)\sin\theta_1\right) \quad (27)$$
Therefore, the $Fos$ against toppling along the $x$ direction, $Fos_{tox}$, results in
$$Fos_{tox} = \frac{M_{stabilizing}}{M_{overturning}} = \frac{M_{W_{inx}} + M_{px}}{M_{W_{outx}} + M_{Hx} \cdot k_1 \cdot k_3 + M_{Ex} \cdot k_2} \quad (28)$$
Similarly, $Fos_{toy}$ can be obtained as
$$Fos_{toy} = \frac{M_{stabilizing}}{M_{overturning}} = \frac{M_{W_{iny}} + M_{py}}{M_{W_{outy}} + M_{Hy} \cdot k_1 + M_{Ey} \cdot k_2} \quad (29)$$
The smaller value is selected as the $Fos$ of the toppling failure mode $Fos_{to}$:
$$Fos_{to} = min(Fos_{tox}, Fos_{toy}) \quad (30)$$
When the stress on mudstone exceeds its strength, it causes partial damage and decreases the stability of the rock block.
Therefore, $Fos$ with the consideration of compressive strength ($Fos_{co}$) and tensional strength ($Fos_{te}$) can be derived as
$$Fos_{co} = \frac{\sigma_{cmax}}{p_{max}} \quad (31)$$
$$Fos_{te} = \frac{\sigma_{tmax}}{-p_{min}} \quad (32)$$
$Fos_{co}$ and $Fos_{te}$ represent the current damage degree of mudstone due to compressive stress and tensile stress, respectively.
When the stress exceeds the ultimate strength, the strength of the mudstone is reduced to the residual value, and the initial
deformation appears. The ability of mudstone to provide resistance to the sliding and toppling of sandstone blocks is thus
reduced, and $Fos_{sl}$ and $Fos_{to}$ subsequently decline. The smaller the value of $Fos_{co}$ and $Fos_{te}$, the greater the damage to the
underlying mudstone. The effective contact area between sandstone and mudstone becomes smaller as the development of
compressive and tension damage, which significantly affects the stability of the overhanging sandstone block.
Finally, four $Fos$ of unstable rock block are obtained. $Fos_{sl}$ and $Fos_{to}$ are routine indicators directly representing the stability
of sandstone blocks. $Fos_{co}$ and $Fos_{te}$ are two indicators proposed in this study for the stability analysis of biased rockfall,
which describe the damage state of the underlying mudstone base. It is necessary to simultaneously consider four $Fos$ to
evaluate the stability of unstable biased rockfall. The entire calculation process is shown in Fig. 9.

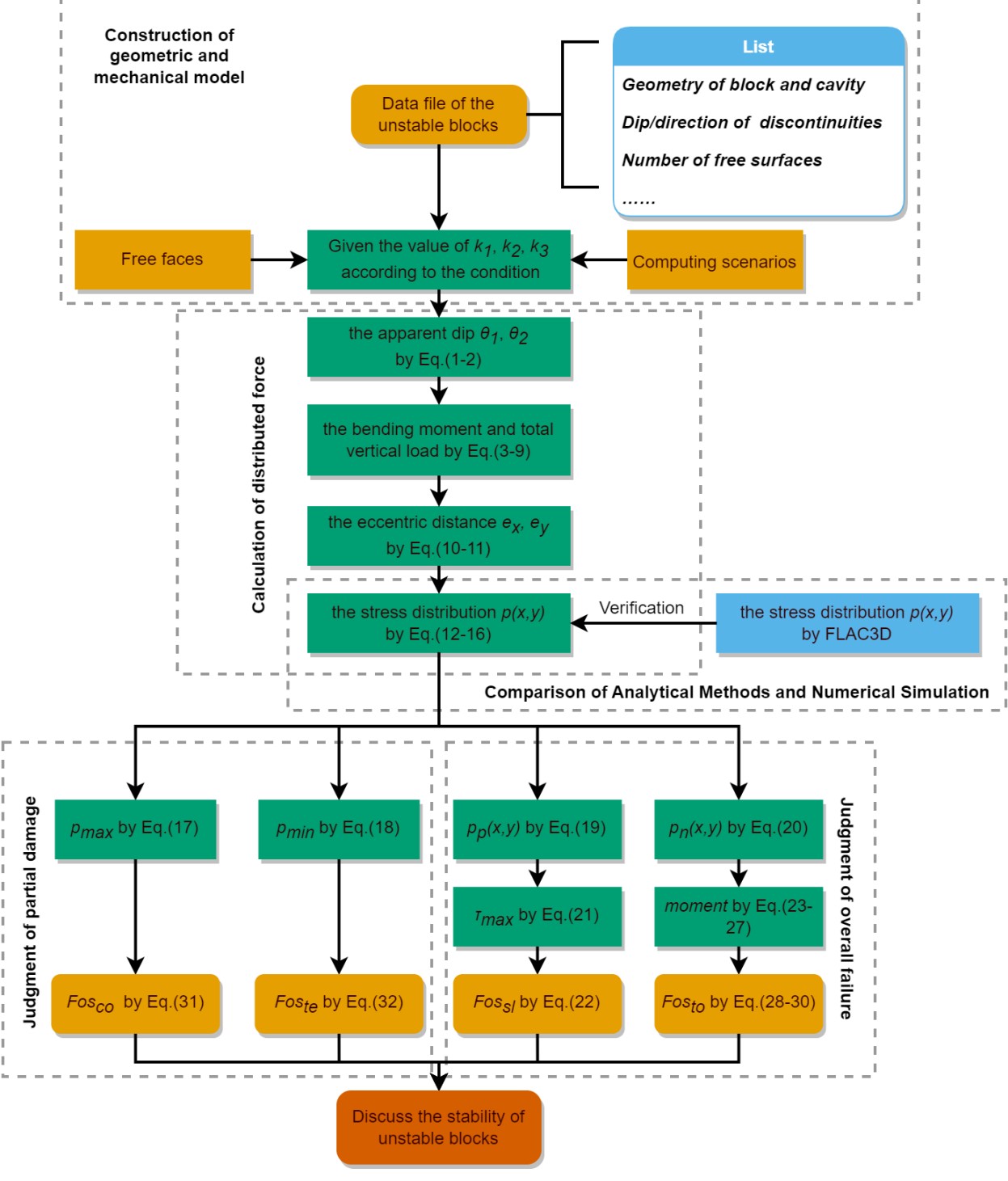


**Figure 9** Calculation process of *Fos* of the unstable rock blocks.
**4 Validation of analytical methods by numerical simulation**
The damage mechanisms at the base of the rock block play an important role in the rockfall evolution process. However, the
stress distribution on the contact surface calculated by the proposed analytical methods is difficult to be validated by the field
data. Therefore, numerical simulation of a biased rockfall was conducted in this study to determine the stress distribution on
the contact surface between overhanging sandstone and underlying mudstone. By comparing the results of the proposed
analytical methods with those obtained from the numerical simulation, the reliability of the analytical methods can be validated.
FLAC3D, a professional software that utilizes the finite difference method (FDM) for three-dimensional analysis of rocks,
soils, and other materials, was employed for the 3D numerical simulation. Based on the geological models, a 3D numerical
simulation model was conducted with FLAC3D 6.00 to analyse the stress distribution on the contact surface (Fig. 10).

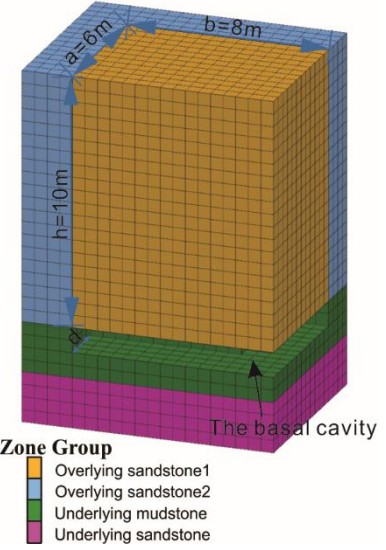


Figure 10 Numerical model built in FLAC3D
The model is mainly composed of sandstone and mudstone, which the Overhanging sandstone1 represents a unstable rock
block (dimensions a×b×h are 6m, 8m, 10m respectively) , and the weathering process of the cavity is represented by excavating
in stages in the underlying mudstone. Sandstone was considered as elastic model, and mudstone was assigned Mohr-Coulomb
model. Material properties were determined by referring to published literature and investigation reports in the study area. The
unit weight of the sandstone block ($\gamma_s$) is 25 kN/m3 (Tang et al., 2010), and the mudstone is 22.54 kN/m3. The friction angle
of the contact surface ($\varphi$) is set to 25° and the cohesion (c) is set to 70 kPa (Zhang et al., 2016). Because of the strength
degradation of mudstone foundations due to intense weathering, the maximum compressive stress of mudstone ($\sigma_{cmax}$) is
replaced by the bearing capacity of mudstone foundations (2300 kPa), which is obtained through plate load tests in adjacent
areas (Zheng et al., 2021). In addition, the maximum tensile stress of mudstone ($\sigma_{tmax}$) is valued as one-ninth of $\sigma_{cmax}$.
The west, north and bottom boundaries of the model are constrained by roller boundary conditions. The cohesion and internal
friction angle of the interface between Overhanging sandstone1 and Overhanging sandstone2 are set to 0. After reaching the
initial force-equilibrium state, the mudstone was excavated to simulate the weathering process, and the vertical stress
distribution on the sand-mudstone interface at different cavity depths was obtained, as shown in Figure 11.

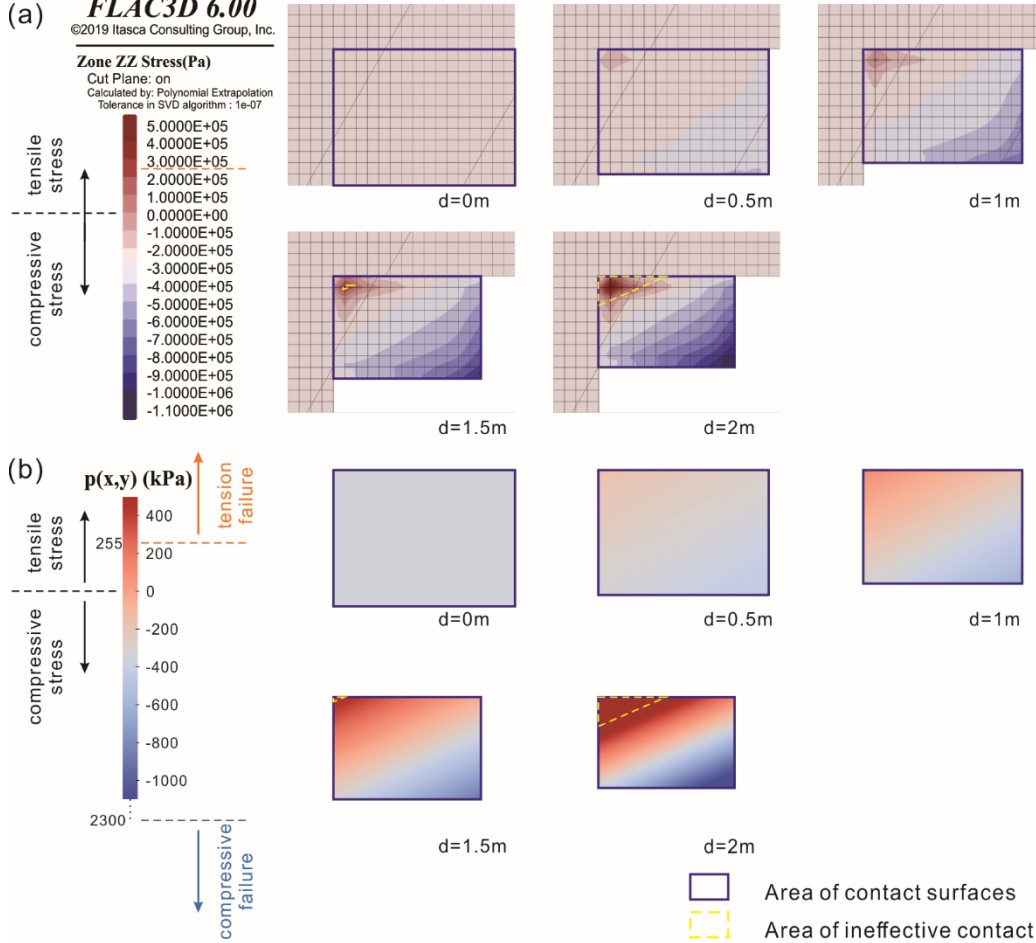

**Fig.11** Diagram of stress distribution in the vertical direction on the contact interface through different methods, (a) the results of numerical simulation by FLAC3D, (b) the results of of proposed analytical method.

When there is no cavity present, represented by d=0m, the stress distribution is uniform compressive stress (According to the
FLAC3D software, compressive stresses are negative). At d=0.5m, the stress remains entirely compressive, but non-uniform
stress distribution occurs on the contact surfaces. At d=1m, the vertical stress value in the upper left corner of the contact
interface surpasses 0 (Fig.11), indicating the presence of tensile stress. As d increases to 1.5m or 2m, the tensile stress in the
upper left corner gradually intensifies, exacerbating the non-uniform stress distribution. The results obtained from the
numerical simulation align with those from the analytical method, confirming the existence of tensile stress at the contact
interface in the biased rockfall due to external erosion development (Fig.11). Tensile stress commonly emerges within the
contact surface, making it challenging to observe directly in the field.

In the context of the limit equilibrium method, the contact area plays a vital role in stability analysis, as shown in Eq. (21)-(30)
in Section 3. The numerical simulation process provides an intuitive understanding of the influence of non-uniform stress
distribution on the contact surfaces on the stability of rock blocks. Whether subjected to tension or compression, the rock layer
has an ultimate strength. In Fig.11, when d=1.5m or 2m, the tensile stress exceeds the ultimate tensile strength, leading to
tensile failure in the upper left corner of the stress distribution diagram. The region enclosed by a yellow dotted line represents
ineffective contact, where no anti-slip force or overturning moment can be generated due to tension failure at the contact
surface. Therefore, this area needs to be subtracted from the total contact area when calculating 〖Fos〗_sl and 〖Fos〗_to.
Similar situations occur when the compressive stress exceeds the ultimate compressive strength. The current maximum
compressive stress has not reached the ultimate compressive strength in Figure 11. However, As d continues to increase, the
area of compression failure will appear in the lower right corner of diagram in Figure 11. This occurrence diminishes the area
capable of providing anti-slip force or overturning moment, thereby reducing the stability of the rock blocks.
The traditional LEM method does not account for distributed forces and fails to consider changes in the contact surface. The
method proposed in this study addresses this issue and is applied to the calculation of the 〖Fos〗_sl and 〖Fos〗_toas
presented in Eq. (21), (25) and (26)).
**5 Results**
A detailed field investigation was carried out in the source area of rockfall (Fig. 3d). The size of the blocks was determined by
on-site measurement with tape and a laser rangefinder. The basal cavities in mudstone were measured with a steel ruler, and
the morphological characteristics of mudstone foundation were mainly described with the average erosion depth of the cavity.
The attitude of discontinuities was measured by compass. The mechanical parameters have been given in Section.4. The height
of the water level ($h_w$) is set to be one-third of $h$, and an earthquake contribution coefficient $k_e$ of 0.05 is considered in stability
calculations. The data obtained from the field survey were organized according to the coordinate system of the geological
model in Section 3.1, and $Fos$ was calculated according to the calculation steps in Section 3.2. The calculated geometric
parameters and $Fos$ results are shown in Table 2.

**Table 2** Geometric parameters of rock blocks in the study area and *Fos* results.

| Block number | Free faces | h [m] | a [m] | b [m] | d₁ [m] | d₂ [m] | d₃ [m] | α [°] | BD | J1 | J2 | $Fos_{te}$ | $Fos_{co}$ | $Fos_{sl}$ | $Fos_{to}$ | $Fos_{min}$ | $Fos_{te}$ | $Fos_{co}$ | $Fos_{sl}$ | $Fos_{to}$ | $Fos_{te}$ | $Fos_{co}$ | $Fos_{sl}$ | $Fos_{to}$ |
|---|---|---|---|---|---|---|---|---|---|---|---|---|---|---|---|---|---|---|---|---|---|---|---|---|
| | | | | | | | | | \multicolumn Dip direction [°] | | | NS (Natural scenario) | | | | | RS (Rainfall scenario) | | | | ES (Earthquake scenario) | | | |
| W01 | 3 | 23 | 7.2 | 6.1 | 0.65 | 0.25 | 0.17 | 6 | 78 | 7 | 97 | - | 2.99 | 5.61 | 101.54 | 2.99 | - | 2.56 | 3.18 | 11.91 | 0.90 | 1.63 | 3.81 | 4.88 |
| W02 | 3 | 23 | 6.42 | 5.25 | 0.78 | 0.4 | 0.31 | 16 | 148 | 51 | 141 | - | 2.84 | 2.10 | 52.28 | 2.10 | - | 2.33 | 1.54 | 8.49 | 0.51 | 1.48 | 1.82 | 3.79 |
| W03 | 2 | 20 | 3.5 | 2.6 | 0.84 | 0.55 | - | 7 | 341 | 53 | 143 | 0.52 | 1.56 | 16.53 | 4.72 | 0.52 | 0.15 | 0.86 | 2.83 | 1.02 | 0.14 | 0.81 | 9.12 | 1.01 |
| W04 | 2 | 19 | 4.6 | 4.6 | 0.62 | 0.77 | - | 7 | 273 | 65 | 155 | 7.35 | 2.37 | - | 24.74 | 2.37 | 0.80 | 1.81 | - | 6.83 | 0.35 | 1.38 | - | 3.23 |
| W05 | 2 | 15 | 16.7 | 5.6 | 2.13 | 1.36 | - | 5 | 283 | 50 | 140 | 1.70 | 2.57 | - | 9.86 | 1.70 | 1.19 | 2.39 | - | 6.10 | 0.63 | 1.99 | - | 3.36 |
| W06 | 3 | 20 | 16.7 | 9.7 | 7.5 | 4.2 | 3.9 | 5 | 302 | 226 | 316 | 0.15 | 0.87 | 8.67 | 1.53 | 0.15 | 0.15 | 0.84 | 4.73 | 1.52 | 0.12 | 0.72 | 5.96 | 1.16 |
| W07 | 2 | 22 | 9.2 | 3.7 | 0.64 | 0.8 | - | 12 | 324 | 315 | 405 | - | 2.27 | 2.82 | 22.86 | 2.27 | 0.57 | 1.55 | 1.62 | 2.97 | 0.34 | 1.28 | 2.44 | 2.21 |
| W08 | 2 | 23 | 12 | 7.9 | 2 | 1.9 | - | 3 | 317 | 332 | 422 | 0.76 | 1.55 | 11.75 | 8.99 | 0.76 | 0.51 | 1.40 | 4.51 | 5.09 | 0.29 | 1.14 | 6.29 | 2.84 |
| W09 | 2 | 18 | 8.4 | 6 | 0.9 | 2.5 | - | 8 | 60 | 335 | 425 | 0.38 | 1.48 | 4.98 | 2.23 | 0.38 | 0.29 | 1.30 | 2.87 | 1.56 | 0.22 | 1.12 | 4.08 | 1.20 |
| W10 | 2 | 23 | 5.7 | 3.3 | 1.3 | 0.85 | - | 5 | 329 | 313 | 403 | 0.30 | 1.16 | 7.41 | 2.53 | 0.30 | 0.12 | 0.71 | 2.30 | 0.71 | 0.11 | 0.68 | 5.84 | 0.75 |
| W11 | 3 | 22 | 1.1 | 2 | 0.1 | 0.64 | 0.1 | 4 | 327 | 120 | 210 | 1.13 | 1.74 | 19.08 | 4.97 | 1.13 | 0.12 | 0.69 | 2.37 | 0.51 | 0.07 | 0.49 | 10.57 | 0.73 |
| W12 | 2 | 25 | 3.9 | 4 | 0.74 | 0.96 | - | 12 | 355 | 297 | 387 | 0.64 | 1.44 | 2.78 | 10.36 | 0.64 | 0.15 | 0.82 | 1.48 | 1.81 | 0.14 | 0.75 | 2.70 | 1.61 |
| W13 | 2 | 12 | 11.9 | 10.9 | 3 | 2.28 | - | 7 | 36 | 73 | 163 | 1.06 | 2.77 | 7.28 | 9.39 | 1.06 | 0.99 | 2.71 | 5.63 | 9.02 | 0.70 | 2.41 | 4.93 | 5.65 |
| W14 | 3 | 19 | 13 | 5 | 0 | 1.1 | 0 | 8 | 296 | 73 | 163 | - | 2.67 | 6.40 | 12.57 | 2.67 | 3.75 | 2.28 | 3.09 | 5.15 | 0.68 | 1.75 | 4.41 | 2.94 |
| W15 | 2 | 18 | 22 | 6 | 8.3 | 0 | - | 8 | 351 | 200 | 290 | 0.70 | 1.84 | 9.74 | 2.93 | 0.70 | 0.60 | 1.75 | 5.03 | 2.83 | 0.39 | 1.50 | 5.79 | 2.34 |
| W16 | 3 | 11 | 5.2 | 7.6 | 0 | 2.9 | 0 | 13 | 42 | 144 | 234 | 1.09 | 3.04 | 3.46 | 3.65 | 1.09 | 1.01 | 2.96 | 2.84 | 3.45 | 0.62 | 2.45 | 2.98 | 2.45 |
| W17 | 3 | 7 | 8 | 2 | 0 | 0.56 | 0 | 20 | 30 | 156 | 246 | 7.71 | 6.72 | 3.07 | 6.83 | 3.07 | 3.40 | 5.87 | 2.29 | 4.49 | 1.48 | 4.70 | 2.81 | 2.86 |
| W18 | 2 | 12 | 8.5 | 4.5 | 1.61 | 1.27 | - | 2 | 252 | 253 | 343 | 0.97 | 2.66 | 20.49 | 7.05 | 0.97 | 0.75 | 2.46 | 10.06 | 4.50 | 0.50 | 2.08 | 8.90 | 2.82 |
| W19 | 2 | 15 | 4.2 | 5.2 | 1.6 | 0.68 | - | 5 | 28 | 56 | 146 | 0.75 | 2.12 | 8.71 | 5.49 | 0.75 | 0.48 | 1.80 | 4.17 | 3.66 | 0.31 | 1.48 | 5.79 | 2.24 |
| W20 | 3 | 15 | 1.8 | 1.7 | 0.23 | 0.5 | 0.3 | 4 | 20 | 63 | 153 | 7.96 | 2.95 | 9.44 | 6.08 | 2.95 | 0.29 | 1.43 | 3.39 | 0.87 | 0.18 | 1.07 | 7.12 | 1.03 |
| W21 | 3 | 20 | 18.9 | 9 | 0 | 2 | 0 | 7 | 348 | 71 | 161 | - | 2.51 | 4.96 | 12.25 | 2.51 | - | 2.36 | 3.31 | 7.48 | 1.15 | 1.90 | 3.58 | 3.95 |
| W22 | 2 | 7 | 5.4 | 5.7 | 1 | 1.65 | - | 6 | 294 | 53 | 143 | 1.53 | 4.48 | - | 5.78 | 1.53 | 1.44 | 4.38 | - | 5.37 | 1.00 | 3.81 | - | 3.88 |

Note: When there is no tensile stress in the mudstone foundation, $Fos_{te}$ has no value. For the case of an anaclinal slope, blocks do not slide and $Fos_{sl}$ has no value. Both parameters are replaced by "-".

## 6 Discussion

### 6.1 Characteristics of rock block stability

There are up to 12 results of $Fos$ per potential unstable block with the consideration of three scenarios and four failure modes (i.e., partial damage and overall failure). Most $Fos_{te}$ values are less than 1 in all scenarios (yellow points in Fig.12), except for two blocks (i.e., W17 and W20), whose $Fos_{te}$ values are also close to 1 under rainfall or earthquake scenarios. Although most of $Fos_{co}$ values (green points in Fig. 12) are greater than 1, they are closer to the critical state of $Fos = 1$ than $Fos_{sl}$ and $Fos_{to}$ (represented by blue and orange points in Fig. 12, respectively). The compression damage of the exposed mudstone can be investigated in the field survey (Fig. 4d). However, it is difficult to observe the phenomenon of tensile damage inside the mudstone base. In the case of weak tensile strength, the mudstone base suffers from tensile failure, and compression failure usually occurs before tension failure. According to the results, their $Fos_{te}$ and $Fos_{co}$ are less than 1 or close to 1, which means that the underlying mudstone has been partially damaged due to slight compressive or tensile failure, and the blocks are potentially unstable with the current depth of the basal cavity. However, most of the blocks do not exhibit overall failure, and they still exist on the slope. Moreover, their $Fos_{sl}$ and $Fos_{to}$ values are greater than 1 in different scenarios, which is consistent with this actuality. The results indicate that most of the blocks are close to a critical state, in which they are partially damaged but the whole block is still stable.

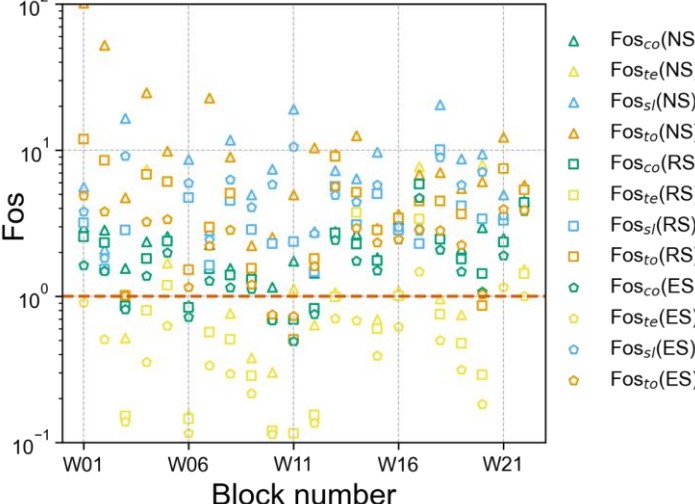

**Figure 12** Distribution of $Fos$ in different scenarios. Shapes represent different scenarios and colours represent different failure modes.

### 6.2 Relationship between $Fos$ and geometric parameters

Fig. 13 presents the relationship between $Fos_{min}$ and two main geometric parameters, the dip of the contact surface and the retreat ratio. In general, the dip angle of the contact surface (α) is the key factor influencing the sliding failure mode. The

horizontal axis in Fig. 13a is α between the rock blocks and underlying mudstone. Most of the points in Fig. 13a are in the
interval [0, 8°], which is consistent with the features of sub-horizontal strata in the study area. The shade of the points does not
change significantly in the $x$-axis direction, as Fig. 13a shows. Therefore, compared with the maximum retreat ratio ($r_{max}$),
the dip of the contact surface has less influence on rockfall stability in the study area. There was a significant positive
correlation between the retreat ratio ($r_{max}$) and $Fos_{min}$. In Fig. 13b, as the retreat ratios increase in the positive direction of
the $x$-axis and $y$-axis, the rock blocks show a notable tendency to be unstable.

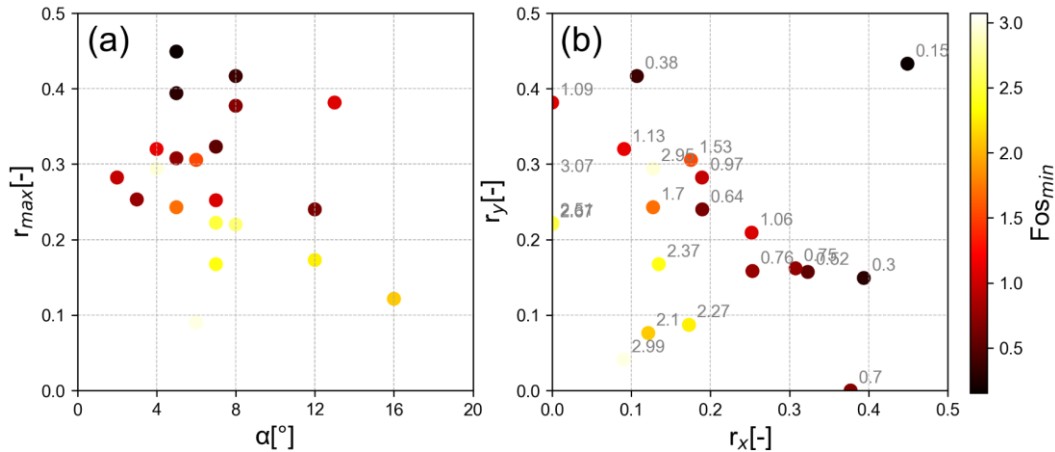


**Figure 13** Correlation between $Fos$ and the dip of contact surface and retreat ratio. Here, α is the dip angle of the contact surface between
rock block and underlaying mudstone, $r_x$ and $r_y$ are the retreat ratio along $x$ direction and $y$ direction, respectively, equal to $d_1/a$ and $d_2/b$,
and $r_{max}$ is the larger of $r_x$ and $r_y$.
**6.3 Definition of rockfall susceptibility**
To explore the variation in $Fos$ with the progressive erosion process of the cavity on the blocks, the cavity retreat velocities in
different directions are assumed to be equal (5 mm/year, Zhang et al. (2016)). Fig. 14 shows the variations in $Fos$ of two
specific blocks during the evolution process of the mudstone cavity. In the initial stage, the cavity is small, and the overhanging
block is stable; all $Fos$ values are greater than 1.0. The cavity expands over time as the mudstone weathers; then, the contact
area decreases, and non-uniform distributed stress arises. When the stress exceeds the ultimate strength of mudstone in a partial
area, $Fos_{co}$ and $Fos_{te}$ decrease significantly, as shown in Fig. 14. The instability of the blocks starts from the failure (or
damage) of the foundation. $Fos_{te}$ and $Fos_{co}$ reach the critical state much earlier than $Fos_{sl}$ and $Fos_{to}$. For these two specific
blocks, when $r_{max}$ increases to 0.4, $Fos_{sl}$ and $Fos_{to}$ are still higher than 1.0. This means that the rock blocks can remain
globally stable in this condition.
These results further elucidate the stability analysis model proposed in this study. $Fos_{co}$ and $Fos_{te}$ introduced in this model
present the damage state of basal mudstone caused by compressive and tensile stresses, which do not provide global instability
of the overhanging block as sliding and toppling. However, $Fos_{co}$ and $Fos_{te}$ are important preliminary signs of subsequent

global failure of the rock block, as presented through the numerical simulation in Section 4. The damage in the basal mudstone can significantly accelerate weathering and prompt expansion of the cavity, which will lead to global failure. The lower $Fos_{co}$ and $Fos_{te}$ are, the lesser the safety margin of the blocks. Therefore, the four $Fos$ used in this study can provide a more comprehensive quantification of rockfall stability.

This result is consistent with Fig. 12, in which 63.7% of the yellow and green points ($Fos_{te}$ and $Fos_{co}$) are located between $Fos = 0.7$ and $Fos = 2.0$. This result can be validated by the field phenomena. In the study area, rock damage (e.g., micro-fractures and cleavages) can be observed in the underlying mudstone. However, most overhanging rock blocks are stable at the present time. This means that even if $Fos_{sl}$ or $Fos_{to}$ is higher than 1, its foundation has begun to be damaged. In the case of heavy rain or earthquakes, $Fos_{sl}$ and $Fos_{to}$ may be reduced to less than 1, and the rockfall occurs.

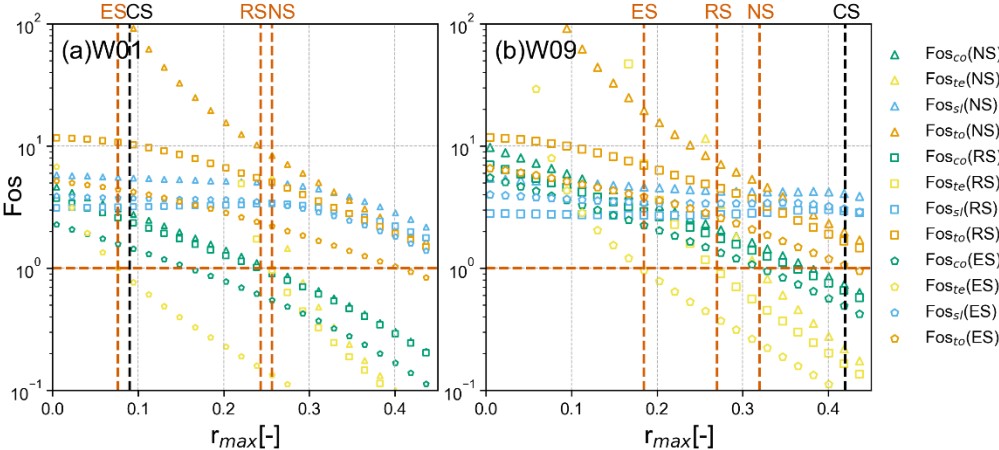

**Figure 14** Variation in $Fos$ with $r_{max}$. (a) and (b) are the results for W01 and W09, respectively, which represent the situation of the blocks with two and three free faces. The black dotted line (CS) approximately represents the current state of the unstable blocks. The red dotted lines correspond to the critical values of $r$ in different scenarios.

Based on the meaning of four $Fos$, rockfall susceptibility can be divided into three levels. When both $Fos_{co}$ and $Fos_{te}$ are greater than 1, the overall rock block is stable, and the mudstone base is not damaged, which is defined as "low susceptibility" and represented by the blue area in Fig. 15. With the development of cavity erosion, when $Fos_{co}$ or $Fos_{te}$ is less than 1 and $Fos_{sl}$ and $Fos_{to}$ are higher than 1, the base undergoes be damaged, and the overhanging sandstone blocks remain relatively stable. This state is defined as "moderate susceptibility" and represented by the yellow area. When $Fos_{sl}$ or $Fos_{to}$ is less than 1 in some scenarios, the rock blocks are in a "high susceptibility" state, which means that rockfalls are highly likely to occur. Fig. 15a indicates that along with the increase in the cavity retreat ratio, the susceptibility of W01 and W09 changes from low susceptibility to moderate susceptibility in the natural scenario. As Fig. 15b and c show, when rainfall or earthquake occurs, $Fos_{sl}$ or $Fos_{to}$ of some blocks is less than 1, which means that some blocks have evolved to the state of high susceptibility and the overall sandstone blocks are unstable.

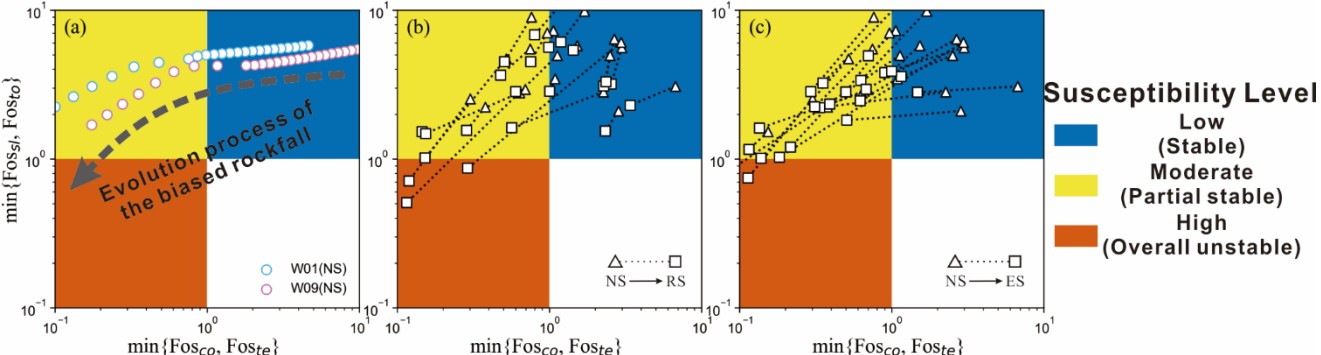

**Figure 15** Rockfall susceptibility based on the combination of four $Fos$. The susceptibility is defined as three levels, represented by red, yellow and blue. (a) shows the progressive failure process of the rock block changing from low susceptibility to moderate susceptibility as the cavity retreat ratio increases (illustrated by W01 and W09 in the natural scenario. (b) and (c) show the change in susceptibility of biased rock blocks, when the scenario changes from natural conditions to rainfall and earthquake conditions.

## 6.4 Critical retreat ratio in the study area

The cavity plays an important role in the progressive failure process of biased rockfall. To analyse the effect of the retreat ratio on the stability of rock blocks, all blocks in the study area were selected to calculate their $Fos$ and susceptibility level with the increasing $r$, whose retreat velocities in different directions are assumed to be equal. Fig. 16 shows that along with the increase in the retreat ratio, the susceptibility level of rock blocks changes from low to moderate susceptibility. Corresponding to the critical state of $\min\{Fos_{co}, Fos_{te}\} = 1$ of all blocks, the minimum retreat ratio is 0.26, and the maximum retreat ratio is 0.41, as marked by the vertical black dotted line in Fig. 16. According to the statistical analysis of critical retreat ratios, both mean and median are 0.33. Therefore, the critical retreat ratio of the rock blocks in the study area can be determined as 0.33, which is marked by the vertical red dotted line in the Fig. 16. The critical retreat ratio calculated by this method can be used for the preliminary identification of potential unstable rock blocks in a specific area, which can help concentrate limited risk treatment resources on these priorities. It should be emphasized that the mechanical parameters and analysis scenarios significantly affect the critical value. Therefore, the elaborative risk control of a given rockfall should be arranged based on its specific parameters and analysis scenarios.

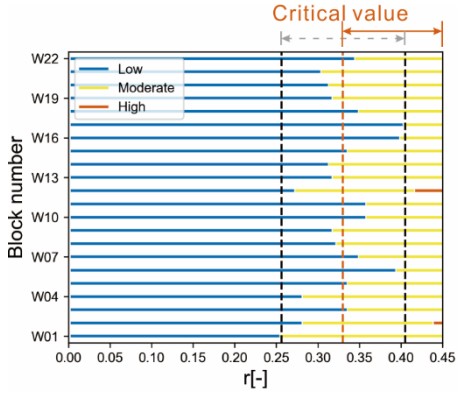


**Figure 16** Effect of the retreat ratio ($r$) on the $Fos$ of the rock block, which is illustrated by all blocks in the study area.
**6.5 Limitations**
This study involves the development of an analytical model for the three-dimensional stability of biased rockfall, combining
the basic LEM method and the consideration of the eccentricity effect. Due to the complexity of rock structure and force
analysis, it is necessary to highlight the limitations of this model.
First, this study uses a three-dimensional coordinate system and bending theory. It is difficult to consider diverse shapes of
rock blocks, and the rock block was simplified as a prismatic column. The assumption of fully persistent discontinuities may
underestimate the stability of rock blocks, and ignores the stress transmission in joints or rock bridges. Then, following the
basic framework of the general LEM method, this study assumed that the rock is not subjected to deformations. The complete
stress–strain behaviour, such as the deformation in the mudstone layer, is not considered in this study. The mode of tension
failure is very difficult to observe in the field, and it is currently verified by means of numerical simulation. Furthermore, the
block stability is strongly influenced by the uncertainty of mechanical parameters. However, because of the difficulties in
sampling strongly weathered mudstone, it is difficult to obtain adequate parameter values for uncertainty statistics. These
limitations will be important considerations in future studies.
**7 Conclusion**
Due to differential weathering in sub-horizontally interbedded of hard rock and soft rock, multi-layer biased rockfalls develop
on steep slopes. In mountainous ranges, cut slopes, and coastal cliffs, rockfall may cause significant facility damage and
casualties in residential areas and transport corridors. The aim of this study was to present a new three-dimensional analytical
method for the stability of rock blocks with basal cavities. In this method, a non-uniform distributed stress due to the
eccentricity effect is applied at the contact surface instead of a point force. The development of non-uniform distributed stress
calculated by the proposed analytical methods was validated by numerical simulation, which presents the evolution process of
biased rockfall from partial damage of the soft underlying layer, caused by non-uniform distributed stress, to toppling and
sliding of overhanging hard rock block due to overall unbalanced force. The method considers four failure modes according
to the rockfall evolution process, including partial damage of the soft foundation ($Fos_{co}$ and $Fos_{te}$) and overall failure of the
rock block ($Fos_{sl}$ and $Fos_{to}$).
Taking the northeast edge of the Sichuan Basin in Southwest China as the study area, the proposed method is used to calculate
the $Fos$ of biased unstable rock blocks. The results show that in the natural scenario, the underlying mudstone of some rock
blocks has been partially damaged, and compression failure of the mudstone has been observed in the field. Some rock blocks
are expected to fail as a whole in rainfall or earthquake scenarios. The statistical analysis indicates that the retreat ratio is the
crucial factor influencing the $Fos$ of biased rockfall. On the basis of different combinations of four $Fos$, rockfall susceptibility
was classified into three levels. As the retreat rate increases, the rock blocks undergo an evolution process from stability to
partial instability and then overall instability. Based on the current mechanical parameters of the eastern Sichuan Basin, the
critical retreat ratio from low to moderate rockfall susceptibility is 0.33.
The proposed method improves the three-dimensional mechanical model of a rock block with a basal cavity by considering
non-uniform distributed stress at the contact surface, which could promote the accuracy of rockfall stability analysis. Due to
the assumptions adopted and the complexity of the failure mechanism of biased rockfall, there are some limitations in this
method, mainly including the simplification of boundary conditions and rock deformation. These limitations will be important
considerations in future studies.
**Data availability**
All raw data can be provided by the corresponding authors upon request.
**Author contributions**
XS, BC and JD planned the campaign; XS and BC performed the field measurements; XS, BC, WW and BL designed and
developed the methodology. XS, BC and JD analysed the data; XS and BC wrote the manuscript draft; JD and WW reviewed
and edited the manuscript.
**Competing interests**
The authors declare that they have no conflicts of interest.

**Acknowledgements**

This research is funded by the National Natural Science Foundation of China (No. 42172318 and No. 42177159). The first author thanks Master Chengjie Luo and Yu Wang for data collection in the field. We also appreciate the assistance of the Research Center of Geohazard Monitoring and Warning in the Three Gorges Reservoir, China.

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
