# Peer review of "A new analytical method for stability analysis of rock blocks with cavities in sub-horizontal strata by the considering eccentricity effect"

_EGUsphere, 2022_

## Author Comment (AC1)

**Answer to RC1**

We earnestly appreciate your time in reviewing the manuscript as well as your valuable comments. Please find our corrections and responses to your comments and suggestions. The corrections are listed in this response and shown in the revised manuscript.

Comments:

● **English language presents some mistakes throughout the paper and needs to be improved.**

Answer:

Thank you for your corrections. We have checked grammar mistakes and improved the wording in the manuscript. Then we further improved the full text according to the comments of reviewer #2.

If the language is still below the journal standard, we will be very glad to hear your comments and suggestions in the future.

● **Figure 8: it is unclear why for case b (three free faces) there is the scheme corresponding to the side view along the x direction (lower and right portion of the figure), since the corresponding face should not exist. Moreover, the upper captions (side view along the x and y directions) should be exchanged, according to this reviewer.**

Answer:

Thank you for your comments. There was ambiguity about the formulation of side view direction in the previous manuscript. The two side views are labeled as *yz* plane and *xz* plane respectively in the new version, and the figure has been corrected.

[Figure]

- **Line 156: the sentence "rainfall is the main predisposing factor of rockfall" is strongly questionable from a theorical point of view. Rainfall is universally known to be not a predisposing factor.**

Answer:

Thank you for your comments. In the universal theorical point of view, rainfall isn't a main predisposing factor of rockfall. However, in the study area, most of the historical rockfall events were recorded after heavy rainfall, because of the hysteretic draining of fissure water due to the obstruction of basal mudstone. The hydrostatic pressure caused by the transient steady flow during heavy rainfall triggers the detachment of rock blocks.

Therefore, in the new version we changed the wording as "According to the historical rockfall events in this area, precipitation is considered as a triggering effect of rock instability."

If it is still ambiguous, we will be very glad to hear your comment in the future. Thank you.

- **The coefficients k in all the equations at pages 11-13 are not introduced at all. Please, check that all the parameters mentioned are clearly defined in the text.**

Answer:

For different scenarios, the three Boolean coefficients enable the formulas to be expressed in a unified form. We explain the role of the three coefficients in the new version as follows. $k_1$, $k_2$ and $k_3$ are the coefficients set to make Eq. 8 and Eq. 9 compatible with different calculation scenarios. So that Eq. 8, Eq. 9 and the following formulas can be expressed in a unified form. At natural scenario, $k_1$ and $k_2$ are both equal to 0. At rainfall scenario, $k_1 = 1$. At earthquake scenario, $k_2 = 1$. For the case of two free faces, $k_3 = 1$. For the case of three free surfaces, $k_3 = 0$.

Besides, we have checked all the parameters in the manuscript to make sure they are clearly defined in the text.

- **The Authors state that, according to the results of in situ surveys, mudstone is not subjected to deformations (line 171). If so, why the need to introduce Fos corresponding to compressive strength (Fos$_{co}$) and tensile strength (Fos$_{te}$). What happens if these strength are reached? What is the effect of stresses exceeding strength in the mudstone? Please, clarify this point, since it represents a central innovative concept proposed in the manuscript, although it is not sufficiently described in detail.**

Answer:

Thank you very much for your professional comments. The expression in initial manuscript about the deformation of mudstone is ambiguous.

According to site survey, compression deformation can be observed in mudstone, which usually manifest as micro-fractures and cleavages. The deformation is very slight and slow in the short term. Therefore, when we analyse the rock block stability in the current state, the deformation of mudstone can be neglected. Besides, $Fos_{sl}$ and $Fos_{to}$ of rock blocks were calculated based on the the limit equilibrium method (LEM). If the deformation of underlying mudstone is considered, the model complexity will be greatly increased. So, in order to reasonably simplify the calculation model, the assumption is proposed that

"mudstone is not subjected to deformations".

Mudstone is loaded by compressive stress and tensile stress. When the stress exceeds the ultimate strength of mudstone, the strength of mudstone is reduced to residual value and the initial deformation appears. As previously mentioned, the deformation is slight, but the micro-fractures and cleavages will accelerate weathering and cause the retreat of cavity. Then, the consequent eccentric effect further increase the compressive stress and tensile stress loading on the mudstone. Therefore, stress exceeding strength in the mudstone will continually accelerate the retreat of mudstone cavity. Mudstone's ability to provide resistance to the sliding and toppling of sandstone blocks will be reduced. $Fos_{sl}$ and $Fos_{to}$ will subsequently decline.

So we introduce $Fos_{co}$ (in the form of the ratio of ultimate compressive strength to maximum compressive stress) to represent the current damage degree of mudstone due to compressive stress. According to the stress distribution pattern of rectangular shape foundation, the stress are redistributed in mudstone. When $Fos_{co} < 1$, it means that the compressive stress of some areas in mudstone exceeds the compressive strength. The partial areas, whose strength have not been exceeded, could provide support to overlying sandstone.

Theoretically, the upper resultant load is placed outside the core of mudstone, tension stress should appear at least in one corner of the mudstone (Fig. 7 Partially unstable state). Therefore, in the same way, we introduced $Fos_{te}$ to represent the damage degree of mudstone due to tension stress. When $Fos_{te} < 1$, it means that the tension stress of some areas exceeds the ultimate tension strength. The smaller the value of $Fos_{co}$ and $Fos_{te}$, the greater the damage to the underlying mudstone. The effective contact area between sandstone and mudstone becomes smaller as the development of compressive and tension damage, which significantly affects the stability of the overlying sandstone block.

If the above explanations are not clear or not adequate, we will be very glad to hear your comments in the future. Thank you very much.

We have modified the text in section 3.1 related to this comment as follows,

"Mudstone is mainly loaded by compressive stress and tensile stress. When the compressive stress of mudstone exceeds its strength in the outer side, the initial damage appears partially. The effective contact surface between mudstone and sandstone is reduced, which aggravate the non-uniform distribution of stress. Therefore, the ability of mudstone providing resistance to the sliding and toppling of overlying sandstone will be reduced.

In the field, compression deformation of mudstone can be observed, which usually manifest as micro-fractures and cleavages. The deformation is very slight and slow in the short term. Besides, the LEM is essentially a Force/Stress approach that do not take into account the deformation. Therefore, in this study, it is assumed that mudstone is not subjected to deformations."

- **Related to the previous point, while the text portion corresponding to the 3D sliding and toppling stability analysis is not new and well-known in the literature, what should be the effect of a $Fos_{co}$ lower than 1.0 from a physical point of view? Is actually important for the block stability? And what about the effect of a $Fos_{te}$ lower than 1.0? The phenomenological and physical interpretation of these**

**concepts seem to be not sufficiently investigated by the Authors.**

Answer:

Thank you for your comment. The effect of $Fos_{co}$ and $Fos_{te}$ lower than 1.0 has been expound in the previous comment. The statement about the phenomenological and physical interpretation of $Fos_{co}$ and $Fos_{te}$ has been added in the new version.

"$Fos_{co}$ and $Fos_{te}$ represent the current damage degree of mudstone due to compressive stress and tensile stress. When the stress exceeds ultimate strength, the strength of mudstone is reduced to residual value, and the initial deformation appears. The ability of mudstone to provide resistance to the sliding and toppling of sandstone blocks will be reduced. $Fos_{sl}$ and $Fos_{to}$ will subsequently decline. The smaller the value of $Fos_{co}$ and $Fos_{te}$, the greater the damage to the underlying mudstone. The effective contact area between sandstone and mudstone becomes smaller as the development of compressive and tension damage, which significantly affects the stability of the overlying sandstone block."

- **It seems that in eq. 31 and 32 the terms should be exchanged: $\sigma_{tmax}$ tmax should refer to $Fos_{te}$, while $\sigma_{cmax}$ refers to $Fos_{co}$. Again, terms $\sigma_{tmax}$ and $\sigma_{cmax}$ have not been defined in the text.**

Answer:

Two formulas were corrected in the new version. The two terms have been defined in the list of symbols, representing the ultimate tensile strength and ultimate compressive strength of mudstone, respectively.

- **Lines 289-290: if there is uncertainty related to the choice of the mechanical parameters, has been such uncertainty quantified? Why not providing a range of the parameter values to account for such uncertainty, along with the corresponding results in terms of Fos?**

Answer:

Thank you very much for your comment. Uncertainty quantification is important for the stability analysis of rockfall. In this manuscript, we mainly focus on the study of stability analysis model based on the traditional limit equilibrium method. Besides, mudstone is difficult to be sampled for laboratory test because of its strong weathering. Field test is an alternative solution, but it is also difficult to obtain adequate parameter values for uncertainty statistics. Therefore, we currently use the results of plate load tests in adjacent area (Zheng et al., 2021). Parameter uncertainty will be an important consideration for us in the future study.

- **Figure 10: if a large amount of cases provides $Fos_{te}$ lower than 1.0, why have the authors not observed tensile failure in the field?**

Answer:

For the same reason as the previous comment, mudstone is difficult to be sampled for laboratory tensile test. So, in this study we valued the tensile strength based on its compression strength with a reduction coefficient of 0.11. According to the calculation results, tensile failure occurs only at a partial area inside the mudstone. Therefore, it is hard to directly observe the internal tensile failures in the field. In addition, partial tensile

failure of mudstone isn't equal to the failure of overlying sandstone. It only means the partial damage of mudstone, which will reduce its resistance to the sliding and toppling of sandstone blocks, and subsequently reduce $Fos_{sl}$ and $Fos_{to}$. Then, for some cases with particularly small $Fos_{te}$ (e.g. W06 $Fos_{te}$ =0.15, W10 $Fos_{te}$ =0.30), the blocks are still stable in the field, we agree with reviewer 2#'s comment "probably because the presence of rock bridges". This is a limitation of our method, which has been added in section 5.5 "Limitations" in the new manuscript.

If the above explanations are not clear or not adequate, we will be very glad to hear your comments in the future. Thank you very much.

We have modified the text in section 5.1 related to this comment as follows,

"The compression failure of the exposed mudstone can be investigated in the field survey (Fig. 4d). However, it is difficult to observe the phenomenon of tensile failure inside the mudstone base. In the case of weak tensile strength, the mudstone base will suffer from tensile failure, and the compression failure usually occurs before the tension failure."

We added a section in the new version to highlight the limitations of the model proposed in this study.

5.5 Limitations

This study proposed an analytical model for three-dimensional stability of biased rockfall, combining the basic LEM method and the consideration of eccentric effect. Due to the complexity of rock structure and force analysis, it is necessary to highlight the limitations of this model.

First, we use a three-dimensional coordinate system and bending theory, it is difficult to consider diverse shapes of rock blocks and complicated fracture water in vertical discontinuities, the rock block was simplified as a prismatic column. The assumption of fully persistent discontinuities may underestimate the stability of rock blocks, it ignores the stress transmission in joints or rock bridges. Then, follow the basic framework of general LEM method, this study assumed that the rock is not subjected to deformations. The complete stress-strain behaviour such as the damage in mudstone layer was not considered in this study. Furthermore, the block stability is strongly influenced by the uncertainty of mechanical parameters. However, because of the difficulties in sampling strong weathered mudstone, it is difficult to obtain adequate parameter values for uncertainty statistics. These limitations will be the important considerations in the future study.

- **Figure 11b does not show rmax, so line 299 is uncorrect. In general, Figure 11b is not adequately explained. Lines 300-301 are uncorrect, since $Fos_{min}$ is not always lower than 1 for the points lying above the red dashed line (see points with 1.53, 2.95, 1.06).**

Answer:

The statement in this part is not rigorous, we have revised in the new version as follows,

"The shade of the points does not change significantly in the x-axis direction as Fig. 11a shows. Therefore, compared with the maximum retreat ratio ($r_{max}$), the dip of contact

surface has fewer influence on rockfall stability in the study area. There is a significant positive correlation between the retreat ratio ($r_{max}$) and $Fos_{min}$. In Fig. 11b, as the retreat ratios increase in the positive direction of the x-axis and y-axis, the rock blocks show an obvious tendency to be unstable."

- **The relationship in Figure 11b (red line) cannot be considered to be generalized for block stability analysis, since the block stability is highly affected by the value of mechanical parameters chosen and the driving factors acting on the block (water level height within the joints, seismic actions), which have been assumed as fixed in the analysis presented. If these input data should vary, the corresponding Fos will change.**

Answer:

Thank you very much for your suggestion. The results in Figure 11b cannot be generalized for block stability analysis. So, we have modified this part concerned with the changing trends of relevant parameters.

"In Fig. 11b, as the retreat ratios increase in the positive direction of the x-axis and y-axis, the rock blocks show an obvious tendency to be unstable."

- **Lines 308-315: this part of the text is highly important because it provides a global interpretation of the conceptual model proposed by the Authors, However, it is excessively synthetic, while it should be enlarged and enriched with a clearer description. The Authors should highlight in a clear way that compressive and tensile states within the block foundation do not provide global instability, as sliding and toppling, but could be only considered as preliminary signs of a possible future failure.**

Answer:

Thank you very much for your professional suggestion. we have revised in the new version as follows,

"Fig.12 shows the variations of $Fos$ of two specific blocks during the evolution process of mudstone cavity. In the initial stage, the cavity is small, and the overlying block is stable, all $Fos$ are greater than 1.0. The cavity expands over time as the mudstone weathers, then the contact area decreases and the non-uniform stress distribution occurs. When the stress exceeds the ultimate strength of mudstone in partial area, $Fos_{co}$ and $Fos_{te}$ decrease significantly as Fig.12 shown. $Fos_{co}$ and $Fos_{te}$ reach critical state much earlier than $Fos_{sl}$ and $Fos_{to}$. To these two specific blocks, when the $r_{max}$ increases to 0.4, $Fos_{sl}$ and $Fos_{to}$ are still higher than 1.0. It means that the rock blocks can still remain global stable in this condition.

These results further elucidate the stability analysis model proposed in this study. $Fos_{co}$ and $Fos_{te}$ introduced in this model present the damage state of basal mudstone caused by compressive and tensile stress, which do not provide global instability of overlying block as sliding and toppling. However, $Fos_{co}$ and $Fos_{te}$ are important preliminary signs of subsequent global failure of rock block. The damage in the basal mudstone could significantly accelerate weathering and prompt expanding of cavity, which will lead to the global failure. The lower the $Fos_{co}$ and $Fos_{te}$, the lesser safety margin the blocks have.

Therefore, the four $Fos$ used in this study could provide a more comprehensive quantification of rockfall stability."

- **Figure 14: this reviewer again strongly suggest to avoid emphasizing excessively the generalization of the results in terms of threshold value for stability, for the same reason described above.**

Answer:

Thank you for the comments about result generalization. In section 5.4, in order to expand the practical significance of this conceptual model, we want to present an analysis method for the critical retreat ratio of potential unstable rock blocks with the same geological structure. Using the samples in the study area, the analysis process was demonstrated. The critical retreat ratio was calculated based on the results at natural scenario (as Figure 14 shows), which can be used for the preliminary identification of potential unstable rock blocks in routine field survey. These identified rock blocks would also be the primary focus when the study area encounters heavy rainfall and earthquake.

Besides, in order to confine the result generalization to specific scenario, we restrict the analysis conclusions to the current study area and further emphasize the influence of mechanical parameters on rockfall stability.

If the above explanations are not clear or not adequate, we will be very glad to hear your comments in the future. Thank you very much.

The results analysis of section 5.4 has been changed as follows,

Fig. 14 shows that along with the increase of retreat ratio, the susceptibility level of rock blocks changes from low to moderate susceptibility. Corresponding to the critical state of $\min\{\text{Fos}_{co}, \text{Fos}_{te}\} = 1$ of all blocks, the minimum retreat ratio is 0.26, and the maximum retreat ratio is 0.41, which are marked by vertical gray dotted line in the Fig. 14. According to the statistics analysis of critical retreat ratios, both the mean and median are 0.33. Therefore, the critical retreat rate of the rock blocks in this study area can be determined as 0.33, which is marked by vertical red dotted line in the Fig. 14.

The critical retreat ratio calculated by this method can be used for the preliminary identification of potential unstable rock blocks in a specific area, which could help concentrating limited risk treatment resources on these priorities. It must be emphasized that the mechanical parameters and analysis scenarios significantly affect the critical value. Therefore, the elaborative risk control of a given rockfall should be arranged based on its specific parameters and analysis scenarios.

- **The Conclusions section**

Answer:

The Conclusions section has been rewritten as follows.

"Due to differential weathering in sub-horizontal layers, multi-layer biased rockfall are developed on the slopes. In mountainous ranges, cut slopes, and coastal cliffs, the rockfall may cause significant facilities damage and casualties in residential areas and transport corridors. The aim of this study was to present a new three-dimensional analytical method for the stability of rock block with basal cavity. A non-uniform distributed force due to eccentric effect was applied at the contact surface, instead of a point force.

Taken the northeast edge of Sichuan basin in Southwest China as study area, the proposed method was used to calculate $Fos$ of the biased unstable rock blocks. The results show that in natural scenario, the underlying mudstone of some rock blocks has been partially damaged, compression failure of the mudstone have been observed in the field. Some rock blocks will fail as a whole in rainfall or earthquake scenarios. The statistical analysis indicates that retreat ratio is the crucial factor influencing the $Fos$ of biased rockfall. On the basis of different critical $Fos$, rockfall susceptibility was classified into three levels. As the retreat rate increases, the rock blocks undergo an evolution process from stability to partial instability and then overall instability. Based on the current mechanical parameters of eastern Sichuan basin, the critical retreat ratio from low to moderate rockfall susceptibility is 0.33.

The proposed method improves the three-dimensional mechanical model of rock block with basal cavity, by considering non-uniform distributed force at the contact surface, which could promote the accuracy of rockfall stability analysis. Due to the assumptions adopted because of the complexity of mechanical failure mechanism of biased rockfall, there are some limitations in this method, mainly including the simplification of boundary conditions and rock deformation. These limitations will be the important considerations in the future study."

---

## Author Comment (AC2)

**Answer to RC2**

We earnestly appreciate your time in reviewing the manuscript as well as your valuable comments. Please find our corrections and responses to your comments and suggestions. The corrections are listed in this response and shown in the revised manuscript.

Comments (Scientific questions):

- **My main concern is that the basic mechanism that you consider is the simplest case when studying the stability of the subsequent blocks close to the cliffs fronts …**

Answer:

Thank you very much for your comments. In this answer, we try to further summarize the innovation and limitation of our model to clarify its basic mechanism. If it is not clear or not adequate, we will be very glad to hear your comments in the future.

This study supplements the basic LEM method with the consideration of eccentric effect. Meanwhile, in order to generalize the basic mechanism of rock blocks with cavity, the model in this study was proposed based on some simplifications.

Firstly, the traditional LEM method only calculates the global stability of rock blocks with natural cavities, which results in overestimation of the stability. Considering the non-uniform stress distribution due to eccentric effect, we introduce partial damage (compressive and tensile damage) of soft underlying layer into LEM.

Besides, since we use a 3D coordinate system and bending theory, it is difficult to consider diverse shapes of rock blocks and complicated fracture water in vertical discontinuities. Therefore, the rock block was simplified as a prismatic column with uniform water height in a fracture. Meanwhile, in the boundary discontinuities of sandstone, rock bridges probably exist to keep stable of rock block. However, the rock bridge is insidious and difficult to be ascertained. So, in this study, we discuss the most adverse state of rock blocks by assuming that the sub-vertical discontinuities has complete connectivity. In the future study, we will improve the basic mechanism of the model by considering complicated rock shape and fracture water state.

- **The next issue is not a limitation only of your method but is a general drawback of the LEM: it does not consider the deformations...**

Answer:

Thank you very much for your comments. This study was putted forward based on the basic assumptions of traditional LEM. Therefore, we don't consider rock deformation.

Besides, in the geological model of this study, there are two kinds of lithology. The sandstone doesn't present distinct deformation before failure because of its high stiffness. Slight deformations can be observed in mudstone before it fails, which usually manifest as rock structure damage, for example micro-fractures and cleavages. The influence of mudstone damage to rock block stability mainly lies in the accelerated weathering, retreat of basal cavity and stress redistribution, rather than the deformation of itself. Therefore, we think it is reasonable to follow the basic assumptions of LEM in this study.

In the text, the statement about rock deformation was not clear and concise. We have

modified in the new version as follows.

"Mudstone is mainly loaded by compressive stress and tensile stress. When the compressive stress of mudstone exceeds its strength in the outer side, the initial damage appears partially. The effective contact surface between mudstone and sandstone is reduced, which aggravate the non-uniform distribution of stress. Therefore, the ability of mudstone providing resistance to the sliding and toppling of overlying sandstone will be reduced.

In the field, compression deformation of mudstone can be observed, which usually manifest as micro-fractures and cleavages. The deformation is very slight and slow in the short term. Besides, the LEM is essentially a Force/Stress approach that do not take into account the deformation. Therefore, in this study, it is assumed that mudstone is not subjected to deformations."

- **Thus, we have to be prudent when examining the results and when deriving conclusions. For instance, in lines 341-342 the authors are discussing some results with four decimal places…**

Answer:

In section 5.4, We have revised this problem.

"Fig. 14 shows that along with the increase of retreat ratio, the susceptibility level of rock blocks changes from low to moderate susceptibility. Corresponding to the critical state of $\min\{\text{Fos}_{co}, \text{Fos}_{te}\} = 1$ of all blocks, the minimum retreat ratio is 0.26, and the maximum retreat ratio is 0.41, which are marked by vertical gray dotted line in the Fig. 14. According to the statistics analysis of critical retreat ratios, both the mean and median are 0.33. Therefore, the critical retreat rate of the rock blocks in this study area can be determined as 0.33, which is marked by vertical red dotted line in the Fig. 14."

- **Another point arises here: Let's consider a 4m wide block with a cavity of 1 m, i.e. retreat ratio of 0.25, stable situation. What will happen if we find a new (or previously hidden) vertical discontinuity in the middle of the block? The retreat ratio changes suddenly to 0.5 and the block becomes unstable. This reasoning highlights the difficulty when trying to use the critical retreat ratio to new sites after the field reconnaissance.**

Answer:

Thank you for this insightful question. Micro-fractures or discontinuities likely form in natural rock blocks. The fully persistent discontinuities may disassemble the former rock block to multiple small ones and change the original stability. We think after field reconnaissance, in each specific site the block stability should be judged based on both critical retreat ratio and elaborative field investigation. The field investigation is supposed to ascertain the boundary condition of rock block at the present time. The random variation of boundary condition isn't easy to be involved in mechanical model.

Besides, inspired by this comment, we added an assumption in this model, "the sandstone block is assumed to be a complete body without persistent discontinuity, and it will not disintegrate before it falls."

- **The Conclusions section must be re-elaborated, now is too short.**

Answer:

The Conclusions section has been rewritten as follows.

"Due to differential weathering in sub-horizontal layers, multi-layer biased rockfall are developed on the slopes. In mountainous ranges, cut slopes, and coastal cliffs, the rockfall may cause significant facilities damage and casualties in residential areas and transport corridors. The aim of this study was to present a new three-dimensional analytical method for the stability of rock block with basal cavity. A non-uniform distributed force due to eccentric effect was applied at the contact surface, instead of a point force.

Taken the northeast edge of Sichuan basin in Southwest China as study area, the proposed method was used to calculate *Fos* of the biased unstable rock blocks. The results show that in natural scenario, the underlying mudstone of some rock blocks has been partially damaged, compression failure of the mudstone have been observed in the field. Some rock blocks will fail as a whole in rainfall or earthquake scenarios. The statistical analysis indicates that retreat ratio is the crucial factor influencing the *Fos* of biased rockfall. On the basis of different critical *Fos*, rockfall susceptibility was classified into three levels. As the retreat rate increases, the rock blocks undergo an evolution process from stability to partial instability and then overall instability. Based on the current mechanical parameters of eastern Sichuan basin, the critical retreat ratio from low to moderate rockfall susceptibility is 0.33.

The proposed method improves the three-dimensional mechanical model of rock block with basal cavity, by considering non-uniform distributed force at the contact surface, which could promote the accuracy of rockfall stability analysis. Due to the assumptions adopted because of the complexity of mechanical failure mechanism of biased rockfall, there are some limitations in this method, mainly including the simplification of boundary conditions and rock deformation. These limitations will be the important considerations in the future study."

- **We have added a section "Limitations" in Discussion.**

Answer:

**5.5 Limitations**

This study proposed an analytical model for three-dimensional stability of biased rockfall, combining the basic LEM method and the consideration of eccentric effect. Due to the complexity of rock structure and force analysis, it is necessary to highlight the limitations of this model.

First, we use a three-dimensional coordinate system and bending theory, it is difficult to consider diverse shapes of rock blocks and complicated fracture water in vertical discontinuities, the rock block was simplified as a prismatic column. The assumption of fully persistent discontinuities may underestimate the stability of rock blocks, it ignores the stress transmission in joints or rock bridges. Then, follow the basic framework of general LEM method, this study assumed that the rock is not subjected to deformations. The complete stress-strain behaviour such as the damage in mudstone layer was not considered in this study. Furthermore, the block stability is strongly influenced by the uncertainty of mechanical parameters. However, because of the difficulties in sampling

strong weathered mudstone, it is difficult to obtain adequate parameter values for uncertainty statistics. These limitations will be the important considerations in the future study.

(Technical corrections)
● **Suggestion: Put all the appearances of *Fos* in italics.**
Answer:
We have corrected the appearances of *Fos* in full text. Thank you very much for all the comments about technical corrections.

● **Line 92: "absence of inventory data" … too sharp to say "absence". Even in your paper, you have some inventory data… I suggest saying "lack of complete inventory data".**
Answer:
"However, its application to rockfall hazards is limited due to the absence of inventory data (Budetta and Nappi, 2013; Malamud et al., 2004)."
**->**
"However, its application to rockfall hazards is limited due to **lack of complete inventory data** (Budetta and Nappi, 2013; Malamud et al., 2004)."

● **Line 100: I guess is "Fig.2c" instead of 2b.**
Answer:
"Frayssines and Hantz (2009) proposed the limit equilibrium method (LEM) to predict block stability considering sliding and toppling in steep limestone cliffs (Fig. 2b)."
**->**
"Frayssines and Hantz (2009) proposed the limit equilibrium method (LEM) to predict block stability considering sliding and toppling in steep limestone cliffs (**Fig. 2c**)."

● **Figure 2a, inset in the graph, "Sagaseta" instead of "Saganseta".**
Answer:
"*Saganseta(1986)*" **->** "***Sagaseta(1986)***"

● **L.110: "to applied" -> "to be applied"**
Answer:
"The supporting force at the contact surface is assumed to applied at a point in the current LEM methods (i.e., N in Fig. 2 b and c)."
**->**
"The supporting force at the contact surface is assumed **to be applied** at a point in the current LEM methods (i.e., N in Fig. 2 b and c)."

● **Fig 3 caption: wording "tectonic sketch profile of A-A' "**
Answer:
"tectonic sketch profile of A-A" -> "**tectonic profile of A-A', whose location is showed in Fig. 3b**".

- **Fig.3 caption: "serial numbers": I think it is not correct. Same in Table 1 columns header.**

Answer:

"serial numbers" -> "**numbers**"

- **\*L. 144: "which" Do you refer to the slopes or to the blocks? "which are consists" wording.**

Answer:

"which" refers to the slopes. The statement was modified to "The slopes in the study area are consist of sub-horizontally interbedding of sandstone and mudstone layers. Therefore, there are multi-layer unstable rock blocks in the slopes."

- **L.80: As is the first appearance of "Eccentric effect", you must define/explain it.**

Answer:

We have added definition of eccentric effect in Introduction.

"Along with the retreat of basal cavity in mudstone layer, the gravity center of the overlying sandstone block moves outward in relation to the mudstone. In this case, the stress distribution in the contact surface of sandstone and mudstone is non-uniform. The mudstone in the outer side bears higher compressive stress than it in the inner side. This phenomenon can be defined as eccentric effect, which will lead to the damage of mudstone and failure of the overlying sandstone by toppling or sliding."

- **L.156, consider using triggering instead of predisposing.**

Answer:

We modify the sentence to "According to the rockfall events in this area, precipitation is the main triggering effect on rock instabilities"

- **Fig.5: lower or upper hemisphere? Which is the location of the data? E1 to E5 show quite different BP dip/dip direction…**

Answer:

The lower hemisphere is marked in new Fig.5. The location of the data is added in the caption of Fig.5 "The data was collected in the rockfall-prone area shown in Fig. 3d." E1 to E5 are all located in sub-horizontal layers. Their BP dips are relatively small. So, their BP dip directions are likely quite different.

- **L.170 "forces" -> "stresses"**

Answer:

"The underlying mudstone plays the role of a rectangular base, which provides non-uniform distributed forces at different locations."

**->**

"The underlying mudstone plays the role of a rectangular base, which provides non-uniform distributed **stresses** at different locations."

- **L183: consider deleting "The predisposing factor's s of". And start the statement: "Rainfall and earthquake …**

Answer:

We have changed the wording "Rainfall and earthquake decrease $Fos$ by generating hydrostatic pressure $H$ in the vertical crack and horizontal seismic force $E$ on the block."

- ***Fig 8: "x direction" must swap with "y direction". "along" is a little ambiguous. Attention: the "z" axis can fall outside the drawings.**

Thank you for this comment. The statement is ambiguous in Fig. 8. In the new version, the two side views are labeled as *yz* plane and *xz* plane, respectively, and Fig. 8 has been corrected. Besides, in Section 3.1, we added a description of coordinate system in Fig.8.

"A Cartesian coordinate system is established in three-dimensional space for the force analysis. The origin O is located at the center of contact surface of sandstone and mudstone. For the case with two free surfaces, the orientation of the free surfaces is set to be the positive direction of x-axis and y-axis, respectively; For the case with three free surfaces, the negative direction of x-axis will also be a free surface. The joint J2 is perpendicular to x-axis, and joint J1 is perpendicular to y-axis."

[Figure]

- **Fig 8 caption: "three free surfaces" -> "three free vertical surfaces"**

Answer:

"(a) and (b) represent the case of unstable rock blocks with two or three free surfaces, respectively."

->

"(a) and (b) represent the case of unstable rock blocks with two or **three free vertical surfaces**, respectively."

● **L189: "Distributed force" … You mean "Stress distribution at the block base"?**

Answer:

"3.2.1 Distributed force"

**->**

"**3.2.1 Stress distribution at the block base**"

● **\*L194: Are you sure of writing "bending moments"? This is not a beam, better saying "non symmetric stress distribution"**

Answer:

bending moments -> non symmetric stress distribution

● **\* Eq. 8 &9: define the factors K1 to k3.**

Answer:

We further explain the role of the three coefficients. For different scenarios, the three Boolean coefficients enable the formulas to be expressed in a unified form.

$k_1$, $k_2$ and $k_3$ are the coefficients set to make Eq. (8) and Eq. (9) compatible with different calculation scenarios. So that Eq. (8), Eq. (9) and the following formulas can be expressed in a unified form. At natural scenario, $k_1$ and $k_2$ are both equal to 0. At rainfall scenario, $k_1 = 1$. At earthquake scenario, $k_2 = 1$. For the case of two free faces, $k_3 = 1$; for the case of three free surfaces, $k_3 = 0$.

● **L229: "underlying" sandstone? Rewrite all the line, please**

Answer:

"$p_p(x, y)$ provides support normal force for the underlying sandstone, and $p_n(x, y)$ provides tension force."

**->**

"$p_p(x, y)$ provides support normal force for the **overlying** sandstone, and $p_n(x, y)$ provides tension force."

● **\*L236: "is not exists"? wording**

Answer:

Added description "For the case of anaclinal slope, the sliding direction is opposite to the free surface. Therefore, the rock block will not slide and $Fos_{sl}$ is not considered in the model."

● **L 258: "aggregate" -> "consider simultaneously"**

Answer:

"It is necessary to aggregate four $Fos$ to judge the stability of unstable rock mass."

**->**

"It is necessary to **consider simultaneously** four $Fos$ to judge the stability of unstable

rock mass."

- **L.264: "…blocks is" -> "blocks was"**

Answer:

"The size of the blocks is determined by on-site measurement with tape and laser rangefinder."

->

"The size of the blocks **was** determined by on-site measurement with tape and laser rangefinder."

- **L266: are ->were**

Answer:

"the morphological characteristics of mudstone foundation are mainly described with the average erosion depth of the cavity."

->

"the morphological characteristics of mudstone foundation were mainly described with the average erosion depth of the cavity."

- **L268: Consider rewriting "are abundantly recorded in the investigation reports and published literatures in this area."**

Answer:

The mechanical parameters of rock blocks were determined referring to the published literature and investigation reports in this area.

- **Table 2: Wording "obtained from the analytical method in section 3"**

Answer:

The title of Table 2 was changed to "Geometric parameters of rock blocks in study area and Fos results"

- **Table 2: consider drawing vertical lines between columns 12 and 13, 17 and 18, and 21 and 22, in order to group the *Fos* by scenarios….**

Answer:

We have added vertical lines between columns 12 and 13, 17 and 18, and 21 and 22 in Table2.

- **L280: Can you improve the section title?**

Answer:

We modified the title of Section 5.1 to "Characteristics of rock block stability".

- **L297: the statement "The shade of the points does not change significantly in the $x$-axis direction, which indicates that the dip of contact surface has little correlation with rockfall stability in this area" seems to me too audacious.**

Answer:

Thank you for your comment. We revised the statement to "The shade of the points does

not change significantly in the x-axis direction as Fig. 11a shows. Therefore, compared with the maximum retreat ratio ($r_{max}$), the dip of contact surface has fewer influence on rockfall stability in the study area."

- **L300: the statement: "$Fos_{min}$ of the points in the upper part are all lower than the critical state ($Fos$ =1)" is false.**

Answer:

Thank you for your comment. It isn't rigorous to divide these points by a straight line. In the new version, we delete this line in Fig. 11b and change the statement as follows.

"In Fig. 11b, as the retreat ratios increase in the positive direction of the x-axis and y-axis, the rock blocks show an obvious tendency to be unstable."

- **Fig. 11 caption: wording**

Answer:

We have modified the caption to "The correlation between $Fos$ and the dip of contact surface and retreat ratio. $\alpha$ is the dip angle of the contact surface between rock block and underlying mudstone. $r_x$ and $r_y$ are the retreat ratio in $x$ direction and $y$ direction, respectively, equal to $d_1/a$ and $d_2/b$. $r_{max}$ is the larger one of $r_x$ and $r_y$."

- **L312: What does it mean "near"? (the vertical axis is Log). L313: Wording: "…well agrees with the field insight, that is most rock blocks…"**

Answer:

We modified this paragraph in the new version.

"Instability of the blocks starts from the failure (or damage) of the foundation. $Fos_{te}$ and $Fos_{co}$ reach critical state much earlier than $Fos_{sl}$ and $Fos_{to}$. This result is consistent with Fig. 10, in which 63.7% of the purple and green points ($Fos_{te}$ and $Fos_{co}$ ) are located between $Fos = 0.7$ and $Fos = 2.0$. This result can be validated by the field phenomena. In the study area, the rock damage (e.g. micro-fractures and cleavages) can be observed in the underlying mudstone. However, most overlying rock blocks are stable at the present time. It means even if $Fos_{sl}$ or $Fos_{to}$ is higher than 1, in fact its foundation has begun to be damaged. In the case of heavy rain or earthquake, $Fos_{sl}$ and $Fos_{to}$ may be reduced to less than 1, and the rockfall will occur."

- **L351: Conclusions. Conclusions section: as stated in the general comments, more stuff must be derived from the study.**

Answer:

We have substantially revised the conclusion section and answered this question above.

---

## Author Response (AR1)

**Answer to RC1**

We earnestly appreciate your time in reviewing the manuscript as well as your valuable comments. Please find our corrections and responses to your comments and suggestions. The corrections are listed in this response and shown in the revised manuscript.

Comments:

● **English language presents some mistakes throughout the paper and needs to be improved.**

Answer:

Thank you for your corrections. We have checked grammar mistakes and improved the wording in the manuscript. Then we further improved the full text according to the comments of reviewer #2.

If the language is still below the journal standard, we will be very glad to hear your comments and suggestions in the future.

● **Figure 8: it is unclear why for case b (three free faces) there is the scheme corresponding to the side view along the x direction (lower and right portion of the figure), since the corresponding face should not exist. Moreover, the upper captions (side view along the x and y directions) should be exchanged, according to this reviewer.**

Answer:

Thank you for your comments. There was ambiguity about the formulation of side view direction in the previous manuscript. The two side views are labeled as *yz* plane and *xz* plane respectively in the new version, and the figure has been corrected.

[Figure]

- **Line 156: the sentence "rainfall is the main predisposing factor of rockfall" is strongly questionable from a theorical point of view. Rainfall is universally known to be not a predisposing factor.**

Answer:

Thank you for your comments. In the universal theorical point of view, rainfall isn't a main predisposing factor of rockfall. However, in the study area, most of the historical rockfall events were recorded after heavy rainfall, because of the hysteretic draining of fissure water due to the obstruction of basal mudstone. The hydrostatic pressure caused by the transient steady flow during heavy rainfall triggers the detachment of rock blocks.

Therefore, in the new version we changed the wording as "According to the historical rockfall events in this area, precipitation is considered a triggering effect of rock instability."

If it is still ambiguous, we will be very glad to hear your comment in the future. Thank you.

- **The coefficients k in all the equations at pages 11-13 are not introduced at all. Please, check that all the parameters mentioned are clearly defined in the text.**

Answer:

For different scenarios, the three Boolean coefficients enable the formulas to be expressed in a unified form. We explain the role of the three coefficients in the new version as follows. "where $k_1$, $k_2$ and $k_3$ are the coefficients set to make Eq. (8) and Eq. (9) compatible with different calculation scenarios. Therefore, Eqs. (8) and (9) and the following formulas can be expressed in a unified form. In the natural scenario, $k_1$ and $k_2$ are both equal to 0. In the rainfall scenario, $k_1 = 1$. In the earthquake scenario, $k_2 = 1$. For the case of two free faces, $k_3 = 1$. For the case of three free surfaces, $k_3 = 0$."

Besides, we have checked all the parameters in the manuscript to make sure they are clearly defined in the text.

- **The Authors state that, according to the results of in situ surveys, mudstone is not subjected to deformations (line 171). If so, why the need to introduce Fos corresponding to compressive strength (Fos$_{co}$) and tensile strength (Fos$_{te}$). What happens if these strength are reached? What is the effect of stresses exceeding strength in the mudstone? Please, clarify this point, since it represents a central innovative concept proposed in the manuscript, although it is not sufficiently described in detail.**

Answer:

Thank you very much for your professional comments. The expression in initial manuscript about the deformation of mudstone is ambiguous.

According to site survey, compression deformation can be observed in mudstone, which usually manifest as micro-fractures and cleavages. The deformation is very slight and slow in the short term. Therefore, when we analyse the rock block stability in the current state, the deformation of mudstone can be neglected. Besides, *Fos$_{sl}$* and *Fos$_{to}$* of rock blocks were calculated based on the the limit equilibrium method (LEM). If the deformation of underlying mudstone is considered, the model complexity will be greatly increased. So, in order to reasonably simplify the calculation model, the assumption is proposed that "mudstone is not subjected to deformations".

Mudstone is loaded by compressive stress and tensile stress. When the stress exceeds the ultimate strength of mudstone, the strength of mudstone is reduced to residual value and the initial deformation appears. As previously mentioned, the deformation is slight, but the micro-fractures and cleavages will accelerate weathering and cause the retreat of cavity. Then, the consequent eccentric effect further increase the compressive stress and tensile stress loading on the mudstone. Therefore, stress exceeding strength in the mudstone will continually accelerate the retreat of mudstone cavity. Mudstone's ability to provide resistance to the sliding and toppling of sandstone blocks will be reduced. $Fos_{sl}$ and $Fos_{to}$ will subsequently decline.

So we introduce $Fos_{co}$ (in the form of the ratio of ultimate compressive strength to maximum compressive stress) to represent the current damage degree of mudstone due to compressive stress. According to the stress distribution pattern of rectangular shape foundation, the stress are redistributed in mudstone. When $Fos_{co} < 1$, it means that the compressive stress of some areas in mudstone exceeds the compressive strength. The partial areas, whose strength have not been exceeded, could provide support to overlying sandstone.

Theoretically, the upper resultant load is placed outside the core of mudstone, tension stress should appear at least in one corner of the mudstone (Fig. 7 Partially unstable state). Therefore, in the same way, we introduced $Fos_{te}$ to represent the damage degree of mudstone due to tension stress. When $Fos_{te} < 1$, it means that the tension stress of some areas exceeds the ultimate tension strength. The smaller the value of $Fos_{co}$ and $Fos_{te}$, the greater the damage to the underlying mudstone. The effective contact area between sandstone and mudstone becomes smaller as the development of compressive and tension damage, which significantly affects the stability of the overlying sandstone block.

If the above explanations are not clear or not adequate, we will be very glad to hear your comments in the future. Thank you very much.

We have modified the text in section 3.1 related to this comment as follows,

"Mudstone is mainly loaded by compressive stress and tensile stress. When the compressive stress of mudstone exceeds its strength on the outer side, some initial damage appears. The effective contact surface between mudstone and sandstone is reduced, which aggravates the non-uniform distribution of stress. In this way, the ability of mudstone to resist the sliding and toppling of overlying sandstone is reduced. In the field, compression deformation of mudstone can be observed, which usually manifests as micro-fractures and cleavages (Fig. 4d). The deformation is very slight and slow in the short term. In addition, the LEM is essentially a force/stress approach that does not take into account the deformation. Therefore, in this study, it is assumed that the mudstone is not subjected to deformation."

- **Related to the previous point, while the text portion corresponding to the 3D sliding and toppling stability analysis is not new and well-known in the literature, what should be the effect of a $Fos_{co}$ lower than 1.0 from a physical point of view? Is actually important for the block stability? And what about the effect of a $Fos_{te}$ lower than 1.0? The phenomenological and physical interpretation of these concepts seem to be not sufficiently investigated by the Authors.**

Answer:

Thank you for your comment. The effect of $Fos_{co}$ and $Fos_{te}$ lower than 1.0 has been expound in the previous comment. The statement about the phenomenological and physical interpretation of $Fos_{co}$ and $Fos_{te}$ has been added in the new version.

"$Fos_{co}$ and $Fos_{te}$ represent the current damage degree of mudstone due to compressive stress and tensile stress, respectively. When the stress exceeds the ultimate strength, the strength of the mudstone is reduced to the residual value, and the initial deformation appears. The ability of mudstone to provide resistance to the sliding and toppling of sandstone blocks is thus reduced, and $Fos_{sl}$ and $Fos_{to}$ subsequently decline. The smaller the value of $Fos_{co}$ and $Fos_{te}$, the greater the damage to the underlying mudstone. The effective contact area between sandstone and mudstone becomes smaller as the development of compressive and tension damage, which significantly affects the stability of the overlying sandstone block."

- **It seems that in eq. 31 and 32 the terms should be exchanged: $\sigma_{tmax}$ tmax should refer to $Fos_{te}$, while $\sigma_{cmax}$ refers to $Fos_{co}$. Again, terms $\sigma_{tmax}$ and $\sigma_{cmax}$ have not been defined in the text.**

Answer:

Two formulas were corrected in the new version. The two terms have been defined in the list of symbols, representing the ultimate tensile strength and ultimate compressive strength of mudstone, respectively.

- **Lines 289-290: if there is uncertainty related to the choice of the mechanical parameters, has been such uncertainty quantified? Why not providing a range of the parameter values to account for such uncertainty, along with the corresponding results in terms of Fos?**

Answer:

Thank you very much for your comment. Uncertainty quantification is important for the stability analysis of rockfall. In this manuscript, we mainly focus on the study of stability analysis model based on the traditional limit equilibrium method. Besides, mudstone is difficult to be sampled for laboratory test because of its strong weathering. Field test is an alternative solution, but it is also difficult to obtain adequate parameter values for uncertainty statistics. Therefore, we currently use the results of plate load tests in adjacent area (Zheng et al., 2021). Parameter uncertainty will be an important consideration for us in the future study.

- **Figure 10: if a large amount of cases provides $Fos_{te}$ lower than 1.0, why have the authors not observed tensile failure in the field?**

Answer:

For the same reason as the previous comment, mudstone is difficult to be sampled for laboratory tensile test. So, in this study we valued the tensile strength based on its compression strength with a reduction coefficient of 0.11. According to the calculation results, tensile failure occurs only at a partial area inside the mudstone. Therefore, it is hard to directly observe the internal tensile failures in the field. In addition, partial tensile failure of mudstone isn't equal to the failure of overlying sandstone. It only means the partial

damage of mudstone, which will reduce its resistance to the sliding and toppling of sandstone blocks, and subsequently reduce $Fos_{sl}$ and $Fos_{to}$. Then, for some cases with particularly small $Fos_{te}$ (e.g. W06 $Fos_{te}$ =0.15, W10 $Fos_{te}$ =0.30), the blocks are still stable in the field, we agree with reviewer 2#'s comment "probably because the presence of rock bridges". This is a limitation of our method, which has been added in section 5.5 "Limitations" in the new manuscript.

If the above explanations are not clear or not adequate, we will be very glad to hear your comments in the future. Thank you very much.

We have modified the text in section 5.1 related to this comment as follows,

"The compression damage of the exposed mudstone can be investigated in the field survey (Fig. 4d). However, it is difficult to observe the phenomenon of tensile damage inside the mudstone base. In the case of weak tensile strength, the mudstone base suffers from tensile failure, and compression failure usually occurs before tension failure."

We added a section in the new version to highlight the limitations of the model proposed in this study.

**5.5 Limitations**

This study involves the development of an analytical model for the three-dimensional stability of biased rockfall, combining the basic LEM method and the consideration of the eccentricity effect. Due to the complexity of rock structure and force analysis, it is necessary to highlight the limitations of this model.

First, this study uses a three-dimensional coordinate system and bending theory. It is difficult to consider diverse shapes of rock blocks, and the rock block was simplified as a prismatic column. The assumption of fully persistent discontinuities may underestimate the stability of rock blocks, and ignores the stress transmission in joints or rock bridges. Then, following the basic framework of the general LEM method, this study assumed that the rock is not subjected to deformations. The complete stress–strain behaviour, such as the deformation in the mudstone layer, is not considered in this study. Furthermore, the block stability is strongly influenced by the uncertainty of mechanical parameters. However, because of the difficulties in sampling strongly weathered mudstone, it is difficult to obtain adequate parameter values for uncertainty statistics. These limitations will be important considerations in future studies.

- **Figure 11b does not show rmax, so line 299 is uncorrect. In general, Figure 11b is not adequately explained. Lines 300-301 are uncorrect, since $Fos_{min}$ is not always lower than 1 for the points lying above the red dashed line (see points with 1.53, 2.95, 1.06).**

Answer:

The statement in this part is not rigorous, we have revised in the new version as follows,

"The shade of the points does not change significantly in the $x$-axis direction, as Fig. 11a shows. Therefore, compared with the maximum retreat ratio ($r_{max}$), the dip of the contact surface has less influence on rockfall stability in the study area. There was a significant positive correlation between the retreat ratio ($r_{max}$) and $Fos_{min}$. In Fig. 11b, as the retreat ratios increase in the positive direction of the $x$-axis and $y$-axis, the rock blocks show a

notable tendency to be unstable."

- **The relationship in Figure 11b (red line) cannot be considered to be generalized for block stability analysis, since the block stability is highly affected by the value of mechanical parameters chosen and the driving factors acting on the block (water level height within the joints, seismic actions), which have been assumed as fixed in the analysis presented. If these input data should vary, the corresponding Fos will change.**

Answer:

Thank you very much for your suggestion. The results in Figure 11b cannot be generalized for block stability analysis. So, we have modified this part concerned with the changing trends of relevant parameters.

"In Fig. 11b, as the retreat ratios increase in the positive direction of the $x$-axis and $y$-axis, the rock blocks show a notable tendency to be unstable."

- **Lines 308-315: this part of the text is highly important because it provides a global interpretation of the conceptual model proposed by the Authors, However, it is excessively synthetic, while it should be enlarged and enriched with a clearer description. The Authors should highlight in a clear way that compressive and tensile states within the block foundation do not provide global instability, as sliding and toppling, but could be only considered as preliminary signs of a possible future failure.**

Answer:

Thank you very much for your professional suggestion. we have revised in the new version as follows,

"Fig. 12 shows the variations in $Fos$ of two specific blocks during the evolution process of the mudstone cavity. In the initial stage, the cavity is small, and the overlying block is stable; all $Fos$ values are greater than 1.0. The cavity expands over time as the mudstone weathers; then, the contact area decreases, and non-uniform distributed stress arises. When the stress exceeds the ultimate strength of mudstone in a partial area, $Fos_{co}$ and $Fos_{te}$ decrease significantly, as shown in Fig. 12. The instability of the blocks starts from the failure (or damage) of the foundation. $Fos_{te}$ and $Fos_{co}$ reach the critical state much earlier than $Fos_{sl}$ and $Fos_{to}$. For these two specific blocks, when $r_{max}$ increases to 0.4, $Fos_{sl}$ and $Fos_{to}$ are still higher than 1.0. This means that the rock blocks can remain globally stable in this condition.

These results further elucidate the stability analysis model proposed in this study. $Fos_{co}$ and $Fos_{te}$ introduced in this model present the damage state of basal mudstone caused by compressive and tensile stresses, which do not provide global instability of the overlying block as sliding and toppling. However, $Fos_{co}$ and $Fos_{te}$ are important preliminary signs of subsequent global failure of the rock block. The damage in the basal mudstone can significantly accelerate weathering and prompt expansion of the cavity, which will lead to global failure. The lower $Fos_{co}$ and $Fos_{te}$ are, the lesser the safety margin of the blocks. Therefore, the four $Fos$ used in this study can provide a more comprehensive quantification of rockfall stability."

- **Figure 14: this reviewer again strongly suggest to avoid emphasizing excessively the generalization of the results in terms of threshold value for stability, for the same reason described above.**

Answer:

Thank you for the comments about result generalization. In section 5.4, in order to expand the practical significance of this conceptual model, we want to present an analysis method for the critical retreat ratio of potential unstable rock blocks with the same geological structure. Using the samples in the study area, the analysis process was demonstrated. The critical retreat ratio was calculated based on the results at natural scenario (as Figure 14 shows), which can be used for the preliminary identification of potential unstable rock blocks in routine field survey. These identified rock blocks would also be the primary focus when the study area encounters heavy rainfall and earthquake.

Besides, in order to confine the result generalization to specific scenario, we restrict the analysis conclusions to the current study area and further emphasize the influence of mechanical parameters on rockfall stability.

If the above explanations are not clear or not adequate, we will be very glad to hear your comments in the future. Thank you very much.

The results analysis of section 5.4 has been changed as follows,

"Fig. 14 shows that along with the increase in the retreat ratio, the susceptibility level of rock blocks changes from low to moderate susceptibility. Corresponding to the critical state of $\min\{Fos_{co}, Fos_{te}\} = 1$ of all blocks, the minimum retreat ratio is 0.26, and the maximum retreat ratio is 0.41, as marked by the vertical grey dotted line in Fig. 14. According to the statistical analysis of critical retreat ratios, both mean and median are 0.33. Therefore, the critical retreat ratio of the rock blocks in the study area can be determined as 0.33, which is marked by the vertical red dotted line in the Fig. 14. The critical retreat ratio calculated by this method can be used for the preliminary identification of potential unstable rock blocks in a specific area, which can help concentrate limited risk treatment resources on these priorities. It should be emphasized that the mechanical parameters and analysis scenarios significantly affect the critical value. Therefore, the elaborative risk control of a given rockfall should be arranged based on its specific parameters and analysis scenarios."

- **The Conclusions section**

Answer:

The Conclusions section has been rewritten as follows.

"Due to differential weathering in sub-horizontally interbedded of hard rock and soft rock, multi-layer biased rockfalls develop on steep slopes. In mountainous ranges, cut slopes, and coastal cliffs, rockfall may cause significant facility damage and casualties in residential areas and transport corridors. The aim of this study was to present a new three-dimensional analytical method for the stability of rock blocks with basal cavities. In this method, a non-uniform distributed stress due to the eccentricity effect is applied at the contact surface instead of a point force. The method considers four failure modes according to the rockfall evolution process, including partial damage of the soft foundation ($Fos_{co}$ and $Fos_{te}$) and overall failure of the rock block ($Fos_{sl}$ and $Fos_{to}$).

Taking the northeast edge of the Sichuan Basin in Southwest China as the study area, the proposed method is used to calculate the $Fos$ of biased unstable rock blocks. The results show that in the natural scenario, the underlying mudstone of some rock blocks has been partially damaged, and compression failure of the mudstone has been observed in the field. Some rock blocks are expected to fail as a whole in rainfall or earthquake scenarios. The statistical analysis indicates that the retreat ratio is the crucial factor influencing the $Fos$ of biased rockfall. On the basis of different combinations of four $Fos$, rockfall susceptibility was classified into three levels. As the retreat rate increases, the rock blocks undergo an evolution process from stability to partial instability and then overall instability. Based on the current mechanical parameters of the eastern Sichuan Basin, the critical retreat ratio from low to moderate rockfall susceptibility is 0.33.

The proposed method improves the three-dimensional mechanical model of a rock block with a basal cavity by considering non-uniform distributed stress at the contact surface, which could promote the accuracy of rockfall stability analysis. Due to the assumptions adopted and the complexity of the failure mechanism of biased rockfall, there are some limitations in this method, mainly including the simplification of boundary conditions and rock deformation. These limitations will be important considerations in future studies."

**Answer to RC2**

We earnestly appreciate your time in reviewing the manuscript as well as your valuable comments. Please find our corrections and responses to your comments and suggestions. The corrections are listed in this response and shown in the revised manuscript.

Comments (Scientific questions):

- **My main concern is that the basic mechanism that you consider is the simplest case when studying the stability of the subsequent blocks close to the cliffs fronts …**

Answer:

Thank you very much for your comments. In this answer, we try to further summarize the innovation and limitation of our model to clarify its basic mechanism. If it is not clear or not adequate, we will be very glad to hear your comments in the future.

This study supplements the basic LEM method with the consideration of eccentric effect. Meanwhile, in order to generalize the basic mechanism of rock blocks with cavity, the model in this study was proposed based on some simplifications.

Firstly, the traditional LEM method only calculates the global stability of rock blocks with natural cavities, which results in overestimation of the stability. Considering the non-uniform stress distribution due to eccentric effect, we introduce partial damage (compressive and tensile damage) of soft underlying layer into LEM.

Besides, since we use a 3D coordinate system and bending theory, it is difficult to consider diverse shapes of rock blocks and complicated fracture water in vertical discontinuities. Therefore, the rock block was simplified as a prismatic column with uniform water height in a fracture. Meanwhile, in the boundary discontinuities of sandstone, rock bridges probably exist to keep stable of rock block. However, the rock bridge is insidious and difficult to be ascertained. So, in this study, we discuss the most adverse state of rock blocks by assuming that the sub-vertical discontinuities has complete connectivity. In the future study, we will improve the basic mechanism of the model by considering complicated rock shape and fracture water state.

- **The next issue is not a limitation only of your method but is a general drawback of the LEM: it does not consider the deformations...**

Answer:

Thank you very much for your comments. This study was putted forward based on the basic assumptions of traditional LEM. Therefore, we don't consider rock deformation.

Besides, in the geological model of this study, there are two kinds of lithology. The sandstone doesn't present distinct deformation before failure because of its high stiffness. Slight deformations can be observed in mudstone before it fails, which usually manifest as rock structure damage, for example micro-fractures and cleavages. The influence of mudstone damage to rock block stability mainly lies in the accelerated weathering, retreat of basal cavity and stress redistribution, rather than the deformation of itself. Therefore, we think it is reasonable to follow the basic assumptions of LEM in this study.

In the text, the statement about rock deformation was not clear and concise. We have

modified in the new version as follows.

"Mudstone is mainly loaded by compressive stress and tensile stress. When the compressive stress of mudstone exceeds its strength on the outer side, some initial damage appears. The effective contact surface between mudstone and sandstone is reduced, which aggravates the non-uniform distribution of stress. In this way, the ability of mudstone to resist the sliding and toppling of overlying sandstone is reduced. In the field, compression deformation of mudstone can be observed, which usually manifests as micro-fractures and cleavages (Fig. 4d). The deformation is very slight and slow in the short term. In addition, the LEM is essentially a force/stress approach that does not take into account the deformation. Therefore, in this study, it is assumed that the mudstone is not subjected to deformation."

- **Thus, we have to be prudent when examining the results and when deriving conclusions. For instance, in lines 341-342 the authors are discussing some results with four decimal places…**

Answer:

In section 5.4, We have revised this problem.

"Fig. 14 shows that along with the increase in the retreat ratio, the susceptibility level of rock blocks changes from low to moderate susceptibility. Corresponding to the critical state of $\min\{Fos_{co}, Fos_{te}\} = 1$ of all blocks, the minimum retreat ratio is 0.26, and the maximum retreat ratio is 0.41, as marked by the vertical grey dotted line in Fig. 14. According to the statistical analysis of critical retreat ratios, both mean and median are 0.33. Therefore, the critical retreat ratio of the rock blocks in the study area can be determined as 0.33, which is marked by the vertical red dotted line in the Fig. 14."

- **Another point arises here: Let's consider a 4m wide block with a cavity of 1 m, i.e. retreat ratio of 0.25, stable situation. What will happen if we find a new (or previously hidden) vertical discontinuity in the middle of the block? The retreat ratio changes suddenly to 0.5 and the block becomes unstable. This reasoning highlights the difficulty when trying to use the critical retreat ratio to new sites after the field reconnaissance.**

Answer:

Thank you for this insightful question. Micro-fractures or discontinuities likely form in natural rock blocks. The fully persistent discontinuities may disassemble the former rock block to multiple small ones and change the original stability. We think after field reconnaissance, in each specific site the block stability should be judged based on both critical retreat ratio and elaborative field investigation. The field investigation is supposed to ascertain the boundary condition of rock block at the present time. The random variation of boundary condition isn't easy to be involved in mechanical model.

Besides, inspired by this comment, we added an assumption in this model, "The sandstone block is assumed to be a complete body without persistent discontinuity, and it will not disintegrate before it falls."

- **The Conclusions section must be re-elaborated, now is too short.**

Answer:

The Conclusions section has been rewritten as follows.

"Due to differential weathering in sub-horizontally interbedded of hard rock and soft rock, multi-layer biased rockfalls develop on steep slopes. In mountainous ranges, cut slopes, and coastal cliffs, rockfall may cause significant facility damage and casualties in residential areas and transport corridors. The aim of this study was to present a new three-dimensional analytical method for the stability of rock blocks with basal cavities. In this method, a non-uniform distributed stress due to the eccentricity effect is applied at the contact surface instead of a point force. The method considers four failure modes according to the rockfall evolution process, including partial damage of the soft foundation ($Fos_{co}$ and $Fos_{te}$) and overall failure of the rock block ($Fos_{sl}$ and $Fos_{to}$).

Taking the northeast edge of the Sichuan Basin in Southwest China as the study area, the proposed method is used to calculate the $Fos$ of biased unstable rock blocks. The results show that in the natural scenario, the underlying mudstone of some rock blocks has been partially damaged, and compression failure of the mudstone has been observed in the field. Some rock blocks are expected to fail as a whole in rainfall or earthquake scenarios. The statistical analysis indicates that the retreat ratio is the crucial factor influencing the $Fos$ of biased rockfall. On the basis of different combinations of four $Fos$, rockfall susceptibility was classified into three levels. As the retreat rate increases, the rock blocks undergo an evolution process from stability to partial instability and then overall instability. Based on the current mechanical parameters of the eastern Sichuan Basin, the critical retreat ratio from low to moderate rockfall susceptibility is 0.33.

The proposed method improves the three-dimensional mechanical model of a rock block with a basal cavity by considering non-uniform distributed stress at the contact surface, which could promote the accuracy of rockfall stability analysis. Due to the assumptions adopted and the complexity of the failure mechanism of biased rockfall, there are some limitations in this method, mainly including the simplification of boundary conditions and rock deformation. These limitations will be important considerations in future studies."

- **We have added a section "Limitations" in Discussion.**

Answer:

**5.5 Limitations**

This study involves the development of an analytical model for the three-dimensional stability of biased rockfall, combining the basic LEM method and the consideration of the eccentricity effect. Due to the complexity of rock structure and force analysis, it is necessary to highlight the limitations of this model.

First, this study uses a three-dimensional coordinate system and bending theory. It is difficult to consider diverse shapes of rock blocks, and the rock block was simplified as a prismatic column. The assumption of fully persistent discontinuities may underestimate the stability of rock blocks, and ignores the stress transmission in joints or rock bridges. Then, following the basic framework of the general LEM method, this study assumed that the rock is not subjected to deformations. The complete stress–strain behaviour, such as the deformation in the mudstone layer, is not considered in this study. Furthermore, the block stability is strongly influenced by the uncertainty of mechanical parameters. However,

because of the difficulties in sampling strongly weathered mudstone, it is difficult to obtain adequate parameter values for uncertainty statistics. These limitations will be important considerations in future studies.

(Technical corrections)
- **Suggestion: Put all the appearances of *Fos* in italics.**

Answer:

We have corrected the appearances of *Fos* in full text. Thank you very much for all the comments about technical corrections.

- **Line 92: "absence of inventory data" … too sharp to say "absence". Even in your paper, you have some inventory data… I suggest saying "lack of complete inventory data".**

Answer:

"However, its application to rockfall hazards is limited due to the absence of inventory data (Budetta and Nappi, 2013; Malamud et al., 2004)."

**->**

"However, its application to rockfall hazards is limited due to **the lack of complete inventory data** (Budetta and Nappi, 2013; Malamud et al., 2004)."

- **Line 100: I guess is "Fig.2c" instead of 2b.**

Answer:

"Frayssines and Hantz (2009) proposed the limit equilibrium method (LEM) to predict block stability considering sliding and toppling in steep limestone cliffs (Fig. 2b)."

**->**

"Frayssines and Hantz (2009) proposed the limit equilibrium method (LEM) to predict block stability considering sliding and toppling in steep limestone cliffs (**Fig. 2c**)."

- **Figure 2a, inset in the graph, "Sagaseta" instead of "Saganseta".**

Answer:

"*Saganseta(1986)*" **->** "***Sagaseta(1986)***"

- **L.110: "to applied" -> "to be applied"**

Answer:

"The supporting force at the contact surface is assumed to applied at a point in the current LEM methods (i.e., N in Fig. 2 b and c)."

**->**

"The supporting force at the contact surface is assumed **to be applied** at a point in the current LEM methods (i.e., N in Fig. 2 b and c)."

- **Fig 3 caption: wording "tectonic sketch profile of A-A' "**

Answer:

"tectonic sketch profile of A-A'" **-> "tectonic sketch profile of A-A', whose location is showed in Fig. 3b;"**

- **Fig.3 caption: "serial numbers": I think it is not correct. Same in Table 1 columns header.**

Answer:

"serial numbers" -> "**numbers**"

- ***L. 144: "which" Do you refer to the slopes or to the blocks? "which are consists" wording.**

Answer:

"which" refers to the slopes. The statement was modified to "The slopes in the study area consist of a sub-horizontally interbedded sandstone and mudstone layer. Therefore, there are multiple layers of potentially unstable rock blocks in the hill slopes (Fig 4a)."

- **L.80: As is the first appearance of "Eccentric effect", you must define/explain it.**

Answer:

We have added definition of eccentric effect in Introduction.

"Along with the retreat of basal cavities in the mudstone layer, the gravity centre of the overlying sandstone block moves outward relative to the mudstone. In this case, the stress distribution in the contact surface of sandstone and mudstone is non-uniform. The mudstone on the outer side bears higher compressive stress than that on the inner side. This phenomenon can be defined as an eccentricity effect, which leads to mudstone damage and failure of the overlying sandstone by toppling or sliding."

- **L.156, consider using triggering instead of predisposing.**

Answer:

We modify the sentence to "According to the historical rockfall events in this area, precipitation is considered a triggering effect of rock instability."

- **Fig.5: lower or upper hemisphere? Which is the location of the data? E1 to E5 show quite different BP dip/dip direction…**

Answer:

The lower hemisphere is marked in new Fig.5. The location of the data is added in the caption of Fig.5 "The data were collected in the rockfall-prone area shown in Fig. 3d." E1 to E5 are all located in sub-horizontal layers. Their BP dips are relatively small. So, their BP dip directions are likely quite different.

- **L.170 "forces" -> "stresses"**

Answer:

"The underlying mudstone plays the role of a rectangular base, which provides non-uniform distributed forces at different locations."

**->**

"The underlying mudstone plays the role of a rectangular base, which provides non-uniform distributed **stresses** at different locations."

- **L183: consider deleting "The predisposing factor's s of". And start the statement: "Rainfall and earthquake …**

Answer:

We have changed the wording "Rainfall and earthquakes decrease $Fos$ by generating hydrostatic pressure $H$ in the vertical crack and horizontal seismic force $E$ on the block."

- **\*Fig 8: "x direction" must swap with "y direction". "along" is a little ambiguous. Attention: the "z" axis can fall outside the drawings.**

Thank you for this comment. The statement is ambiguous in Fig. 8. In the new version, the two side views are labeled as *yz* plane and *xz* plane, respectively, and Fig. 8 has been corrected. Besides, in Section 3.1, we added a description of coordinate system in Fig.8.

"A Cartesian coordinate system is established in three-dimensional space for the force analysis. The origin $O$ is located at the centre of the contact surface between sandstone and mudstone. For the case with two free surfaces, the orientation of the free surfaces is set to be the positive direction of the $x$-axis and $y$-axis. For the case with three free surfaces, the negative direction of the $x$-axis is also a free surface. Joint J2 is perpendicular to the $x$-axis, and joint J1 is perpendicular to the $y$-axis."

- **Fig 8 caption: "three free surfaces" -> "three free vertical surfaces"**

Answer:

"(a) and (b) represent the case of unstable rock blocks with two or three free surfaces, respectively."

->

"(a) and (b) represent the case of unstable rock blocks with two or **three free vertical surfaces**, respectively."

● **L189: "Distributed force" … You mean "Stress distribution at the block base"?**
Answer:
"3.2.1 Distributed force"
**->**
"**3.2.1 Stress distribution at the block base**"

● **\*L194: Are you sure of writing "bending moments"? This is not a beam, better saying "non symmetric stress distribution"**
Answer:

bending moments -> **non-symmetrical stress distribution**

● **\* Eq. 8 &9: define the factors K1 to k3.**
Answer:
We further explain the role of the three coefficients. For different scenarios, the three Boolean coefficients enable the formulas to be expressed in a unified form.
"where $k_1$, $k_2$ and $k_3$ are the coefficients set to make Eq. (8) and Eq. (9) compatible with different calculation scenarios. Therefore, Eqs. (8) and (9) and the following formulas can be expressed in a unified form. In the natural scenario, $k_1$ and $k_2$ are both equal to 0. In the rainfall scenario, $k_1 = 1$. In the earthquake scenario, $k_2 = 1$. For the case of two free faces, $k_3 = 1$. For the case of three free surfaces, $k_3 = 0$."

● **L229: "underlying" sandstone? Rewrite all the line, please**
Answer:
"$p_p(x, y)$ provides support normal force for the underlying sandstone, and $p_n(x, y)$ provides tension force."
**->**
"Here, $p_p(x, y)$ provides support normal force for the **overlying** sandstone, and $p_n(x, y)$ provides tension force."

● **\*L236: "is not exists"? wording**
Answer:
Added description "For the case of an anaclinal slope, the sliding direction is opposite to the free surface. Therefore, the rock block does not slide, and $Fos_{sl}$ is not considered in the model."

● **L 258: "aggregate" -> "consider simultaneously"**
Answer:
"It is necessary to aggregate four $Fos$ to judge the stability of unstable rock mass."
**->**
"It is necessary to **simultaneously consider** four $Fos$ to evaluate the stability of unstable

biased rockfall."

- **L.264: "…blocks is" -> "blocks was"**

Answer:

"The size of the blocks is determined by on-site measurement with tape and laser rangefinder."

**->**

"The size of the blocks **was** determined by on-site measurement with tape and a laser rangefinder."

- **L266: are ->were**

Answer:

"the morphological characteristics of mudstone foundation are mainly described with the average erosion depth of the cavity."

**->**

"and the morphological characteristics of mudstone foundation were mainly described with the average erosion depth of the cavity."

- **L268: Consider rewriting "are abundantly recorded in the investigation reports and published literatures in this area."**

Answer:

"The mechanical parameters for the $Fos$ calculation of rock blocks were determined by referring to published literature and investigation reports in this area."

- **Table 2: Wording "obtained from the analytical method in section 3"**

Answer:

The title of Table 2 was changed to "Geometric parameters of rock blocks in the study area and $Fos$ results."

- **Table 2: consider drawing vertical lines between columns 12 and 13, 17 and 18, and 21 and 22, in order to group the *Fos* by scenarios….**

Answer:

We have added vertical lines between columns 12 and 13, 17 and 18, and 21 and 22 in Table2.

- **L280: Can you improve the section title?**

Answer:

We modified the title of Section 5.1 to "Characteristics of rock block stability".

- **L297: the statement "The shade of the points does not change significantly in the $x$-axis direction, which indicates that the dip of contact surface has little correlation with rockfall stability in this area" seems to me too audacious.**

Answer:

Thank you for your comment. We revised the statement to "The shade of the points does

not change significantly in the $x$-axis direction, as Fig. 11a shows. Therefore, compared with the maximum retreat ratio ($r_{max}$), the dip of the contact surface has less influence on rockfall stability in the study area."

- **L300: the statement: "$Fos_{min}$ of the points in the upper part are all lower than the critical state ($Fos$ =1)" is false.**

Answer:

Thank you for your comment. It isn't rigorous to divide these points by a straight line. In the new version, we delete this line in Fig. 11b and change the statement as follows.

"In Fig. 11b, as the retreat ratios increase in the positive direction of the $x$-axis and $y$-axis, the rock blocks show a notable tendency to be unstable."

- **Fig. 11 caption: wording**

Answer:

We have modified the caption to "Correlation between $Fos$ and the dip of contact surface and retreat ratio. Here, $\alpha$ is the dip angle of the contact surface between rock block and underlaying mudstone, $r_x$ and $r_y$ are the retreat ratio along $x$ direction and $y$ direction, respectively, equal to $d_1/a$ and $d_2/b$, and $r_{max}$ is the larger of $r_x$ and $r_y$."

- **L312: What does it mean "near"? (the vertical axis is Log). L313: Wording: "…well agrees with the field insight, that is most rock blocks…"**

Answer:

We modified this paragraph in the new version.

"The instability of the blocks starts from the failure (or damage) of the foundation. $Fos_{te}$ and $Fos_{co}$ reach the critical state much earlier than $Fos_{sl}$ and $Fos_{to}$. For these two specific blocks, when $r_{max}$ increases to 0.4, $Fos_{sl}$ and $Fos_{to}$ are still higher than 1.0. This means that the rock blocks can remain globally stable in this condition.

These results further elucidate the stability analysis model proposed in this study. $Fos_{co}$ and $Fos_{te}$ introduced in this model present the damage state of basal mudstone caused by compressive and tensile stresses, which do not provide global instability of the overlying block as sliding and toppling. However, $Fos_{co}$ and $Fos_{te}$ are important preliminary signs of subsequent global failure of the rock block. The damage in the basal mudstone can significantly accelerate weathering and prompt expansion of the cavity, which will lead to global failure. The lower $Fos_{co}$ and $Fos_{te}$ are, the lesser the safety margin of the blocks. Therefore, the four $Fos$ used in this study can provide a more comprehensive quantification of rockfall stability.

This result is consistent with Fig. 10, in which 63.7% of the purple and green points ($Fos_{te}$ and $Fos_{co}$) are located between $Fos = 0.7$ and $Fos = 2.0$. This result can be validated by the field phenomena. In the study area, rock damage (e.g., micro-fractures and cleavages) can be observed in the underlying mudstone. However, most overlying rock blocks are stable at the present time. This means that even if $Fos_{sl}$ or $Fos_{to}$ is higher than 1, its foundation has begun to be damaged. In the case of heavy rain or earthquakes, $Fos_{sl}$ and $Fos_{to}$ may be reduced to less than 1, and the rockfall occurs.

- **L351: Conclusions. Conclusions section: as stated in the general comments, more stuff must be derived from the study.**

Answer:

We have substantially revised the conclusion section and answered this question above.

---

## Author Response (AR2)

**Answer to EC:**

We earnestly appreciate your time in reviewing the manuscript as well as your valuable comments. Please find our corrections and responses to your comments and suggestions. The corrections are listed in this response and shown in the revised manuscript.

Comments:

- **I think that the quality of the paper should be strongly improved and that several scientific steps are only inferred and not described accurately. The referee #1 pointed out: "The deformation and failure mechanisms described in the manuscript are only inferred by the authors based on field observations, but these mechanisms are not corroborated by in-situ evidences, monitoring or, at least, numerical modelling, which could be potentially highly useful to such specific purposes."**

Answer:

In previous manuscripts, the failure mechanisms was not clearly described. The stress distribution on the contact surface is difficult to be observed in reality, and there is currently no effective monitoring data. We especially thank reviewer 1 for his suggestion to carry out the related comparison work by numerical simulation. For the stress distribution, the results of numerical simulation through FLAC3D are consistent with the results of the proposed analytical method, which verifies the validity and rationality of the analytical method. A new Section has been added to the article, entitled " 4 Validation of analytical methods by numerical simulation".

The damage mechanisms at the base of the rock block play an important role in the rockfall evolution process. However, the stress distribution on the contact surface calculated by the proposed analytical methods is difficult to be validated by the field data. Therefore, numerical simulation of a biased rockfall was conducted in this study to determine the stress distribution on the contact surface between overhanging sandstone and underlying mudstone. By comparing the results of the proposed analytical methods with those obtained from the numerical simulation, the reliability of the analytical methods can be validated. FLAC3D, a professional software that utilizes the finite difference method (FDM) for three-dimensional analysis of rocks, soils, and other materials, was employed for the 3D numerical simulation. Based on the geological models, a 3D numerical simulation model was conducted with FLAC3D 6.00 to analyse the stress distribution on the contact surface (Fig. 10).

[Figure]

Figure 10 Numerical model built in FLAC3D

The model is mainly composed of sandstone and mudstone, which the Overhanging sandstone1 represents a unstable rock block (dimensions $a \times b \times h$ are 6m, 8m, 10m respectively) , and the weathering process of the cavity is represented by excavating in stages in the underlying mudstone. Sandstone was considered as elastic model, and mudstone was assigned Mohr-Coulomb model. Material properties were determined by referring to published literature and investigation reports in the study area. The unit weight of the sandstone block ($\gamma_s$) is 25 kN/m3 (Tang et al., 2010), and the mudstone is 22.54 kN/m3. The friction angle of the contact surface ($\varphi$) is set to 25° and the cohesion (c) is set to 70 kPa (Zhang et al., 2016). Because of the strength degradation of mudstone foundations due to intense weathering, the maximum compressive stress of mudstone ($\sigma_{cmax}$) is replaced by the bearing capacity of mudstone foundations (2300 kPa), which is obtained through plate load tests in adjacent areas (Zheng et al., 2021). In addition, the maximum tensile stress of mudstone ($\sigma_{tmax}$) is valued as one-ninth of $\sigma_{cmax}$. The west, north and bottom boundaries of the model are constrained by roller boundary conditions. The cohesion and internal friction angle of the interface between Overhanging sandstone1 and Overhanging sandstone2 are set to 0. After reaching the initial force-equilibrium state, the mudstone was excavated to simulate the weathering process, and the vertical stress distribution on the sand-mudstone interface at different cavity depths was obtained, as shown in Figure 11.

[Figure]

Fig.11 Diagram of stress distribution in the vertical direction on the contact interface through different methods, (a) the results of numerical simulation by FLAC3D, (b) the results of   of proposed analytical method.

When there is no cavity present, represented by d=0m, the stress distribution is uniform compressive stress (According to the FLAC3D software, compressive stresses are negative).  At d=0.5m, the stress remains entirely compressive, but non-uniform stress distribution occurs on the contact surfaces. At d=1m, the vertical stress value in the upper left corner of the contact interface surpasses 0 (Fig.11), indicating the presence of tensile stress. As d increases to 1.5m or 2m, the tensile stress in the upper left corner gradually intensifies, exacerbating the non-uniform stress distribution. The results obtained from the numerical simulation align with those from the analytical method, confirming the existence of tensile stress at the contact interface in the biased rockfall due to external erosion development (Fig.11). Tensile stress commonly emerges within the contact surface, making it challenging to observe directly in the field.

- **The referee #1 also pointed out an important shortcoming "Since the damage mechanisms at the base of the rock blocks play an important role in the geological context described, according to the authors, they should try to demonstrate the same mechanisms at some extent. The relationship existing between damage in the underlying mudstone and the block stability in terms of**

**toppling and sliding mechanisms is only inferred, while the consequences are taken into account in the analytical model proposed. Such relationship should be investigated more in detail."**

Answer:

The concept of ineffective contact surface is proposed in Fig. 11. With the development of differential weathering and non-uniform distributed stress, the underlying mudstone is damaged due to tensile and compressive stress in excess of strength, and the area of ineffective contact surface increases. The area that can provide anti-slip force and overturning moment is reduced, the overlying blocks are more prone to failure by sliding or toppling. The relevant content is expressed in Section.4.

In the context of the limit equilibrium method, the contact area plays a vital role in stability analysis, as shown in Eq. (21)-(30) in Section 3. The numerical simulation process provides an intuitive understanding of the influence of non-uniform stress distribution on the contact surfaces on the stability of rock blocks. Whether subjected to tension or compression, the rock layer has an ultimate strength. In Fig.11, when d=1.5m or 2m, the tensile stress exceeds the ultimate tensile strength, leading to tensile failure in the upper left corner of the stress distribution diagram. The region enclosed by a yellow dotted line represents ineffective contact, where no anti-slip force or overturning moment can be generated due to tension failure at the contact surface. Therefore, this area needs to be subtracted from the total contact area when calculating $\mathrm{Fos_{sl}}$ and $\mathrm{Fos_{to}}$. Similar situations occur when the compressive stress exceeds the ultimate compressive strength. The current maximum compressive stress has not reached the ultimate compressive strength in Figure 11. However, As $d$ continues to increase, the area of compression failure will appear in the lower right corner of diagram in Figure 11. This occurrence diminishes the area capable of providing anti-slip force or overturning moment, thereby reducing the stability of the rock blocks.

The traditional LEM method does not account for distributed forces and fails to consider changes in the contact surface. The method proposed in this study addresses this issue and is applied to the calculation of the $\mathrm{Fos_{sl}}$ and $\mathrm{Fos_{to}}$ as presented in Eq. (21), (25) and (26)).

- **The referee #1**
  Figure 7: cavity is not a correct word to be used in the description of the specific geological situation (cavity is a void "inside" a rock mass); external erosion should work better; the overlying block should be called "overhanging".

Answer:

The content of Figure 7 has been modified. The overlying block has been changed to overhanging block in the manuscript.

[Figure]

- **The referee #1**

  Line 272: "according to the principle of friction" is not a rigorous expression; Eq. 21 represents the mathematical formulation of the Mohr-Coulomb criterion.

Answer:

According to the principle of friction, the ultimate shear strength →

According to **the Mohr-Coulomb criterion**, the ultimate shear strength

In addition, the abstract, introduction, conclusion and other parts of the article have been revised. Please refer to the revised manuscript for related information.